# Offline Model-Based Optimization: Comprehensive Review

**Minsu Kim**[*]                                              *minsu.kim@mila.quebec*
*Mila - Quebec AI Institute/KAIST*

**Jiayao (Claris) Gu**[*]                                     *jia.yao.gu@mail.mcgill.ca*
*Mila - Quebec AI Institute/McGill University*

**Ye Yuan**                                                   *ye.yuan3@mail.mcgill.ca*
*Mila - Quebec AI Institute/McGill University*

**Taeyoung Yun**                                              *taeyoung.yun@mila.quebec*
*Mila - Quebec AI Institute/KAIST*

**Zixuan Liu**                                                *zucksliu@cs.washington.edu*
*University of Washington*

**Yoshua Bengio**                                             *yoshua.bengio@mila.quebec*
*Mila - Quebec AI Institute/University of Montreal*

**Can (Sam) Chen**[†]                                         *can.chen@mila.quebec*
*Mila - Quebec AI Institute/University of Montreal*

**Reviewed on OpenReview:** *https://openreview.net/forum?id=QcSZWo1TLl*

## Abstract

*Offline black-box optimization* is a fundamental challenge in science and engineering, where the goal is to optimize black-box functions using only offline datasets. This setting is particularly relevant when querying the objective function is prohibitively expensive or infeasible, with applications spanning protein engineering, material discovery, neural architecture search, and beyond. The main difficulty lies in accurately estimating the objective landscape beyond the available data, where extrapolations are fraught with significant epistemic uncertainty. This uncertainty can lead to *objective hacking* (*reward hacking*)—exploiting model inaccuracies in unseen regions—or other spurious optimizations that yield misleadingly high performance estimates outside the offline distribution. Recent advances in *model-based optimization* (MBO) have harnessed the generalization capabilities of deep neural networks to develop offline-specific surrogate and generative models. Trained with carefully designed strategies, these models are more robust against out-of-distribution issues, facilitating the discovery of improved designs. Despite its growing impact in accelerating scientific discovery, the field lacks a comprehensive review. To bridge this gap, we present the first thorough review of offline MBO. We begin by formalizing the problem for both *single-objective* and *multi-objective* settings and by reviewing recent benchmarks and evaluation metrics. We then categorize existing approaches into two key areas: *surrogate modeling*, which emphasizes accurate function approximation in out-of-distribution regions, and *generative modeling*, which explores high-dimensional design spaces to identify high-performing designs. Finally, we examine the key challenges and propose promising directions for advancement in this rapidly evolving field including safe control of superintelligent systems. For a curated list of resources, please visit our repository.

---

*[*]Minsu and Claris contributed equally to this work.
*[†]Correspondence: can.chen@mila.quebec or chencan421@gmail.com

# 1 Introduction

*Offline black-box optimization* is a fundamental challenge in science and engineering, where the objective is to optimize a black-box function using only a fixed dataset (Trabucco et al., 2022). This setting has broad applications, including protein engineering (Sarkisyan et al., 2016; Yang et al., 2019), material discovery (Hamidieh, 2018), and neural architecture search (Lu et al., 2023). For instance, in neural architecture search, the goal is to identify high-performing architectures solely from existing architecture-performance pairs, without training any new models which can be expensive. Unlike *online black-box optimization*, which allows direct interaction with the objective function, offline black-box optimization is particularly relevant when querying the function is costly, time-consuming, or infeasible (Angermüller et al., 2020; Barrera et al., 2016; Sample et al., 2019).

Offline black-box optimization is challenging because it requires accurately estimating the landscape of the black-box function beyond the available offline data (Trabucco et al., 2021). Extrapolating into these unseen regions suffers from significant epistemic uncertainty, arising from limited data coverage and imperfect model knowledge rather than inherent noise (Der Kiureghian & Ditlevsen, 2009), which manifests in two problematic scenarios. On the one hand, it may trigger *reward hacking* (Skalse et al., 2022) (which we refer to as *objective hacking* in our text), wherein the model exploits inaccuracies within the objective estimation in regions beyond offline data. On the other hand, it can give rise to other forms of spurious optimization – especially in guided generative modeling where the model overfits spurious correlations within offline data – that yield misleadingly high-performance sample generation outside offline distribution (Brookes et al., 2019).

Offline *model-based optimization* (MBO) is a paradigm that leverages the generalization capabilities of deep neural networks to develop offline-specific surrogate and generative models for solving offline black-box optimization problems. This progress has spurred two complementary lines of research. One line focuses on building *surrogate models* that extrapolate beyond the offline dataset, enabling robust function approximation and reliable gradient-based optimization to improve existing designs (Trabucco et al., 2021; Fu & Levine, 2021). The other line explores the use of *generative models* to navigate high-dimensional design spaces more effectively, facilitating the discovery of high-performing designs underrepresented in the offline data (Kumar & Levine, 2020; Kim et al., 2024b). Importantly, these two lines are not mutually exclusive – surrogate and generative models often complement each other to enhance overall performance (Fannjiang & Listgarten, 2020; Chen et al., 2024). The design-centric offline MBO differs from offline RL (Levine et al., 2020; Prudencio et al., 2023), which seeks to learn an optimal policy from trajectory data and emphasizes sequential decision-making and credit assignment in Markov decision processes, whereas offline MBO focuses on single-step design generation from static datasets.

Despite the rapid progress in offline MBO, both newcomers and seasoned researchers find it challenging to stay abreast of its evolving methodologies. Furthermore, the diversity of approaches and objectives has led to a fragmented landscape, making it difficult to discern overarching trends. To address these challenges, we present the first comprehensive review on offline MBO, synthesizing recent advances, categorizing key areas, and highlighting emerging directions. This review serves as both an accessible introduction for newcomers and a structured synthesis for experts looking to navigate the evolving frontiers of offline MBO.

In this work, we formalize the problem settings for both offline *single-* and *multi-objective optimization* (Section 2). We also introduce a generative modeling perspective that frames offline black-box optimization as *conditional generation*, compare single- and multi-objective settings, and discuss the connections between online and offline black-box optimization. Additionally, we review recent benchmarks and propose a taxonomy that categorizes them into four application areas (Section 3): (1) *synthetic function*, (2) *real-world system*, (3) *scientific design*, and (4) *machine learning (ML) model*. Evaluation costs tend to increase – and our understanding of the underlying mechanisms tends to decrease – from categories (1) through (3). We address category (4) separately, given its growing prominence in the ML community. For each category, we detail the associated tasks, including the number of objectives and the oracle evaluators. We also provide an overview of the commonly used evaluation metrics, including *usefulness*, *novelty*, and *diversity*.

Next, we categorize existing approaches into two key research lines—as we have discussed above: *surrogate modeling* (Section 4) and *generative modeling* (Section 5). Importantly, these approaches are not mutually

exclusive, and surrogate models and generative models are often used together in offline MBO. We separate their discussion in this review to provide clearer insights into each component, while also exploring in detail how they interact and complement one another. Finally, we conclude our paper (Section 6) by outlining promising future directions in this rapidly evolving field. In particular, we highlight several key areas for further exploration: (1) *robust and realistic benchmarking*, (2) *uncertainty estimation of surrogate models*, (3) *causal graphical surrogate models*, (4) *advanced generative modeling*, (5) and *application to LLM alignment and AI Safety.* An outline of our paper organization is depicted in Figure 1.

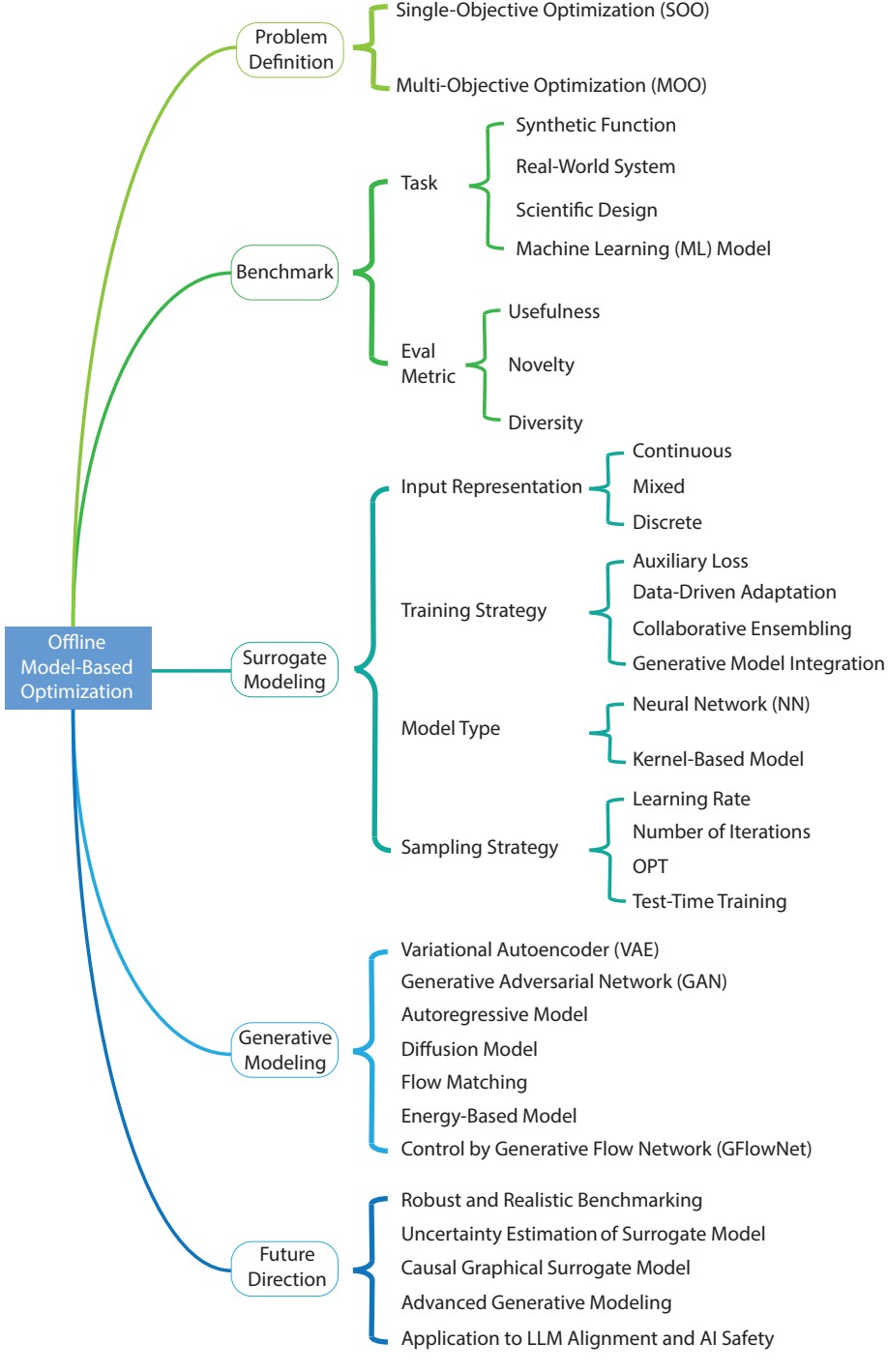

Figure 1: Paper overview.

## 2   Problem Definition

In offline black-box optimization, the goal is to discover a new design, denoted by $\boldsymbol{x}^*$, that maximizes the objective(s) $\boldsymbol{f}(\boldsymbol{x})$. This is achieved using an offline dataset $\mathcal{D}$, which consists of $N$ designs paired with their property labels. In particular, the dataset is given by

$$\mathcal{D} = \{(\boldsymbol{x}_i, \boldsymbol{y}_i)\}_{i=1}^N \tag{1}$$

where each design vector $\boldsymbol{x}_i$ belongs to a design space $\mathcal{X} \subseteq \mathbb{R}^d$, and each property label $\boldsymbol{y}_i \in \mathbb{R}^m$ contains the corresponding $m$ objective values for that design. The function $\boldsymbol{f} : \mathcal{X} \to \mathbb{R}^m$ maps a design to its $m$-dimensional objective value vector. Offline MBO applies to static optimization problems where the objective $\boldsymbol{f}(\boldsymbol{x})$ is a fixed black-box function without temporal structure, distinguishing it from sequential decision-making problems such as system identification followed by optimal control (Sutton & Barto, 1998).

**Single-Objective Optimization**   In offline *single-objective optimization* (SOO), only one objective is considered (i.e., $m = 1$), leading to the formulation:

$$\boldsymbol{x}^* = \arg\max_{\boldsymbol{x}} f(\boldsymbol{x}) \,.$$

For instance, the design $\boldsymbol{x}$ might represent a neural network architecture, with $f(\boldsymbol{x})$ denoting the network's accuracy on a given dataset (Zoph & Le, 2017).

A prevalent method for addressing this problem involves training a deep neural network (DNN) surrogate model, $f_{\boldsymbol{\phi}}(\cdot)$, with parameters $\boldsymbol{\phi}$ on an offline dataset using supervised learning. The model parameters are optimized by minimizing the mean squared error between the model's predictions and the true labels:

$$\boldsymbol{\phi}^* = \arg\min_{\boldsymbol{\phi}} \frac{1}{N} \sum_{i=1}^N \left( f_{\boldsymbol{\phi}}(\boldsymbol{x}_i) - y_i \right)^2 . \tag{2}$$

After training, the surrogate $f_{\boldsymbol{\phi}^*}(\cdot)$ is employed as a stand-in for the true objective function, and design optimization proceeds via gradient ascent updates:

$$\boldsymbol{x}_{t+1} = \boldsymbol{x}_t + \eta \nabla_{\boldsymbol{x}} f_{\boldsymbol{\phi}^*}(\boldsymbol{x}) \Big|_{\boldsymbol{x}=\boldsymbol{x}_t}, \quad \text{for } t \in [0, T-1], \tag{3}$$

where $\eta$ is the learning rate and $T$ is the total number of iterations. The final design, $\boldsymbol{x}_T$, is then taken as the candidate solution.

A critical challenge of this method is the accurate estimation of the objective landscape beyond the region covered by the offline data. In these extrapolated regions, the surrogate's posterior predictive variance captures its *epistemic uncertainty*—the component due to limited data and model knowledge rather than inherent noise. Formally, if we place a parameter posterior $p(\boldsymbol{\theta} \mid \mathcal{D})$ over model parameters, then the predictive distribution under this posterior is

$$p(y \mid \boldsymbol{x}, \mathcal{D}) = \int p(y \mid \boldsymbol{x}, \boldsymbol{\theta}) \, p(\boldsymbol{\theta} \mid \mathcal{D}) \, d\boldsymbol{\theta},$$

whose total variance decomposes into

$$\mathrm{Var}[y \mid \boldsymbol{x}, \mathcal{D}] = \underbrace{\mathbb{E}_{p(\boldsymbol{\theta}|\mathcal{D})}\big[\mathrm{Var}(y \mid \boldsymbol{x}, \boldsymbol{\theta})\big]}_{\text{aleatoric}} + \underbrace{\mathrm{Var}_{p(\boldsymbol{\theta}|\mathcal{D})}\big[\mathbb{E}(y \mid \boldsymbol{x}, \boldsymbol{\theta})\big]}_{\text{epistemic}},$$

where the first term quantifies aleatoric uncertainty arising from inherent noise in the data, and the second term quantifies epistemic uncertainty and diminishes as more data are observed (Der Kiureghian & Ditlevsen, 2009). In practice, epistemic uncertainty can be estimated via:

- **Gaussian Processes**: posterior variance gives closed-form decomposition (Rasmussen & Williams, 2006).

- **Deep Ensembles**: variance across multiple network predictions approximates parameter uncertainty (Lakshminarayanan et al., 2017).

- **Monte Carlo Dropout**: repeated stochastic forward passes approximate the weight posterior (Gal & Ghahramani, 2016).

- **Bayesian Neural Networks**: variational or sampling-based inference yields predictive distributions with parameter uncertainty.

High epistemic uncertainty in out-of-distribution regions can lead to *objective hacking*, where the optimizer exploits surrogate inaccuracies to identify spurious high-performing designs. We discuss strategies for building robust surrogates in Section 4.

**Multi-Objective Optimization** Offline *multi-objective optimization* (MOO) extends the framework to simultaneously address multiple objectives using the dataset $\mathcal{D}$. In this setting, the goal is to find solutions that balance competing objectives effectively. For instance, when designing a neural architecture, one might seek to achieve both high accuracy and high efficiency (Lu et al., 2023). Formally, the multi-objective optimization problem is defined as:

$$\text{Find } \boldsymbol{x}^* \in \mathcal{X} \text{ such that there is no } \boldsymbol{x} \in \mathcal{X} \text{ with } \boldsymbol{f}(\boldsymbol{x}) \succ \boldsymbol{f}(\boldsymbol{x}^*), \tag{4}$$

where $\boldsymbol{f} : \mathcal{X} \to \mathbb{R}^m$ is the vector of $m$ objective functions and the symbol $\succ$ indicates *Pareto dominance*. Specifically, a solution $\boldsymbol{x}$ is said to *Pareto dominate* another solution $\boldsymbol{x}^*$ (denoted $\boldsymbol{f}(\boldsymbol{x}) \succ \boldsymbol{f}(\boldsymbol{x}^*)$) if

$$\forall i \in \{1, \ldots, m\}, \quad f_i(\boldsymbol{x}) \geq f_i(\boldsymbol{x}^*) \quad \text{and} \quad \exists j \in \{1, \ldots, m\} \text{ such that } f_j(\boldsymbol{x}) > f_j(\boldsymbol{x}^*). \tag{5}$$

In simpler terms, $\boldsymbol{x}$ is no worse than $\boldsymbol{x}^*$ in every objective and is strictly better in at least one. A design is considered *Pareto optimal* if no other design in $\mathcal{X}$ Pareto dominates it. The collection of all such Pareto optimal designs forms the *Pareto set* (PS), and the corresponding set of objective vectors,

$$\{\boldsymbol{f}(\boldsymbol{x}) \mid \boldsymbol{x} \in \text{PS}\}, \tag{6}$$

is known as the *Pareto front* (PF). The overarching aim in MOO is to obtain a diverse set of solutions that closely approximates the PF, thereby capturing the best possible trade-offs among the objectives. Analogous to the single-objective case, a naive approach involves modeling each of the $m$ objectives with separate surrogate models and combining their predictions through a weighted sum to compute the gradient (Ma et al., 2020). However, this approach fails to account for conflicts among objectives and is also susceptible to *objective hacking*. We discuss approaches for addressing these issues in Section 4.

**Generative Modeling** In addition to the surrogate modeling discussed above, generative modeling is another key ingredient in offline MBO. In fact, offline MBO methods can be viewed through the lens of *conditional generation*, where the objective is to model the distribution $p(\boldsymbol{x} \mid \boldsymbol{y}_c)$ with $\boldsymbol{y}_c$ representing the desired conditions. By applying Bayes' rule, this distribution can be decomposed as

$$p(\boldsymbol{x} \mid \boldsymbol{y}_c) \propto p(\boldsymbol{x})\, p(\boldsymbol{y}_c \mid \boldsymbol{x}). \tag{7}$$

From this perspective, offline MBO methods generally fall into two categories: *inverse* and *forward*. Inverse methods directly train a conditional generative model for $p(\boldsymbol{x} \mid \boldsymbol{y}_c)$. For example, MIN (Brookes et al., 2019) employs the generative adversarial network (GAN)-based inverse mapping from $\boldsymbol{y}$ to $\boldsymbol{x}$ to generate designs that meet the desired specifications. Forward methods, on the other hand, leverage a surrogate model for $p(\boldsymbol{y}_c \mid \boldsymbol{x})$ to guide an unconditional generative model $p(\boldsymbol{x})$. For instance, ROMA (Yu et al., 2021a) computes gradients in the latent space of a variational autoencoder (VAE) to iteratively refine generated designs. Similarly, gradient-based approaches such as conservative objective models (COMs) (Trabucco et al., 2021) and ICT (Yuan et al., 2023) fall into this category, although many of these methods do not explicitly model the generative component $p(\boldsymbol{x})$.

Since the literature on offline MBO often integrates surrogate and generative modeling, we separate their discussion in this review to provide clearer insights into each component: *surrogate modeling* is discussed in Section 4, while *generative modeling* is covered in Section 5. We discuss generative modeling here only at a high level to provide a novel conditional-generation view of offline MBO, and revisit it in Section 5 for a detailed exposition of methodological developments.

**Comparison between SOO and MOO**   Both single-objective optimization (SOO) and multi-objective optimization (MOO) strive to optimize objectives using only an offline dataset, which leads to some inherent similarities. In both settings, surrogate models are built to approximate the objective function(s), and generative models are employed to explore the design space efficiently.

However, significant methodological differences emerge due to their distinct optimization goals. One major distinction lies in the training of surrogate models. In SOO, the surrogate is typically trained to minimize the prediction error for a single objective while incorporating relevant priors, thereby facilitating direct gradient-based optimization. In contrast, MOO must capture the interdependencies among multiple objectives, often by leveraging their relationships to enhance surrogate modeling. This challenge usually necessitates the use of multi-task learning strategies (Chen et al., 2018; Yu et al., 2020).

Another fundamental difference is the sampling strategy. In SOO, the focus is on a single property–such as model accuracy in neural architecture design–which simplifies conditional generation to either computing the gradient of that property (Chen et al., 2023a) or building an inverse mapping from the property to the design (Kumar & Levine, 2020). These approaches, however, are not directly applicable to MOO, where improvements in one objective (e.g., accuracy) might lead to the deterioration of another (e.g., efficiency). Consequently, MOO relies on *Pareto-aware* sampling strategies to navigate the trade-offs among conflicting objectives. For example, Paretoflow (Yuan et al., 2025) assigns uniform weight vectors to different objectives to guide flow matching towards the Pareto front.

In summary, while SOO and MOO share foundational principles and both contend with out-of-distribution issues, MOO introduces additional challenges related to balancing competing objectives, thereby requiring specialized modeling and optimization strategies.

**Relation with Online Black-Box Optimization.**   *online black-box optimization* involves iteratively querying an expensive ground-truth objective function for new design points and then using these new data points to update a surrogate model (Jain et al., 2022; Gruver et al., 2023; Frey et al., 2025). In high-cost domains such as drug discovery—where each experiment (e.g., a clinical trial) can be prohibitively expensive (Angermüller et al., 2020)—these queries must be made sparingly under a limited budget. A central challenge in online black-box optimization is balancing the trade-off between *exploitation*—selecting designs with high predicted objective values—and *exploration*—querying uncertain regions to improve the model's understanding of the objective landscape. This exploration-exploitation dilemma has been widely studied in the context of Bayesian optimization (Frazier, 2018), where acquisition functions such as the Upper Confidence Bound and Expected Improvement are commonly used to guide query selection. This principle has also been explored in the operations research, notably through Efficient Global Optimization (EGO) (Jones et al., 1998).

Bayesian models, such as Gaussian Processes (GPs) (Williams & Rasmussen, 1995; Rasmussen & Williams, 2006), are commonly used for online black-box optimization because they provide principled *uncertainty quantification*. However, GPs have cubic complexity in the number of data points, which becomes infeasible for large datasets. As a result, deep neural networks equipped with approximate uncertainty estimation techniques (e.g., Monte Carlo dropout or ensembles) are often employed, though their uncertainty estimates can be less reliable in practice (Gal & Ghahramani, 2016; Lakshminarayanan et al., 2017).

In offline black-box optimization, there is no ability to query the objective after an initial dataset has been collected—effectively a *zero-round* version of online black-box optimization. Since no further data can be acquired, offline methods often avoid high-uncertainty regions where the surrogate model might be highly inaccurate or risky. Consequently, offline black-box optimization algorithms often adopt more conservative strategies, prioritizing regions of the design space that are both high reward and well-understood according

to the existing dataset. One example from online black-box optimization is the use of the Upper Confidence Bound (Frazier, 2018), an acquisition function that promotes exploration by combining predicted reward with uncertainty. By adopting a Lower Confidence Bound (Frazier, 2018), another acquisition function that promotes exploitation and favors high-reward but low-uncertainty regions, the same principle can be adapted for offline MBO. This contrast illustrates the core difference between online and offline black-box optimization: exploration versus conservatism.

Despite these differing strategies—online seeking to reduce uncertainty versus offline avoiding it—both paradigms rely on accurate and scalable uncertainty quantification. Scalable methods for uncertainty estimation can thus benefit both fields. We highlight two candidate methods: Neural Processes, which circumvent the cubic complexity of GPs (Garnelo et al., 2018), and GFlowNets, which can offer amortized Bayesian posterior inference over the parameter space to estimate model uncertainty (Bengio et al., 2023).

Futhermore online and offline methods can synergize naturally. In the online phase, the goal is to collect a high-quality dataset by judiciously querying promising and uncertain regions. Once the budget for queries is exhausted and the dataset becomes static, offline black-box optimization can then leverage the collected data for a final decision. In this way, online black-box optimization focuses on acquiring the most informative dataset, while offline black-box optimization focuses on extracting the best solution from the available data.

## 3 Benchmark

In this section, we present a systematic overview of benchmarks in offline MBO by first categorizing the available tasks in Section 3.1 and then discussing the evaluation metrics in Section 3.2.

### 3.1 Task

We begin by grouping tasks into four main categories: (1) *synthetic function*, which leverages closed-form mathematical functions to provide efficient, scalable, and analytically transparent benchmarks (see Section 3.1.1); (2) *real-world system*, which addresses practical engineering challenges in domains such as robotics (see Section 3.1.2); (3) *scientific design*, encompassing applications in biology, chemistry, and material science (see Section 3.1.3); and (4) *machine learning model*, which includes problems like neural architecture design (see Section 3.1.4). Evaluation costs tend to rise—and our understanding of the underlying mechanisms tends to diminish—as we move from category (1) to category (3). We address category (4) separately, reflecting its increasing significance within the ML community.

It is important to note that these categories are not mutually exclusive; some benchmarks may belong to more than one category. For example, Ehrlich (Stanton et al., 2024) can be viewed as a *synthetic function* task for biological sequence optimization, yet we mainly discuss it under *scientific design* to better reflect its application context. Each subsubsection first discusses the intrinsic advantages and limitations of its task category and then introduces the commonly used tasks within that category. For each task, we report its *size* (i.e., the offline dataset), *space* (design space), *dimension* (design dimension), *# Obj* (number of objectives), and *oracle* (evaluation oracle type). This organization offers a balanced view of both the theoretical appeal and practical applicability of each task category.

#### 3.1.1 Synthetic Function

*Synthetic function* tasks use closed-form mathematical functions to generate offline datasets for offline MBO methods. During evaluation, these same functions serve as oracles to benchmark these methods. These functions can, in principle, also be applied in an online context; however, in this discussion they serve as benchmarks in the offline setting, where algorithms are limited to using only the offline dataset.

**Advantages** (1) *Computational efficiency*: With closed-form expressions, these functions are quick to compute, allowing for extensive experimentation. (2) *Scalability*: Their ability to be defined for arbitrary input dimensions and numbers of objectives makes them highly adaptable for evaluating algorithms on large-scale problems. (3) *Analytical transparency*: The known analytical forms enable exact computation

of the Pareto front along with other key properties like gradients and Hessians, which helps in thoroughly understanding both the problem landscape and the behavior of optimization algorithms.

**Limitations** (1) *Lack of realism*: The inherent simplicity and often smooth nature of synthetic functions can fail to capture the complexities of real-world problems, including discontinuous or highly constrained landscapes. For example, in protein sequence design, a single amino acid change can cause a dramatic, discontinuous shift in properties, and the design space itself is highly constrained by biological functionality (Angermüller et al., 2020; Jain et al., 2022). Consequently, performance on synthetic tasks may not directly translate to practical applications. (2) *Limited adaptability for deep learning*: Deep neural networks have achieved significant success in areas such as image (Krizhevsky et al., 2012), language (Brown et al., 2020), and molecule (Jumper et al., 2021), and recent work has extended these models to offline MBO (Chen et al., 2023b; Watson et al., 2023). However, synthetic data generated by mathematical functions often lack the rich, complex patterns found in real-world data (e.g., molecular structures), which prevents deep learning methods from fully demonstrating their potential in these benchmarks.

Below, we introduce specific synthetic function tasks for both single-objective and multi-objective optimization, covering most tasks from commonly used benchmarks such as the BayesO benchmark (Kim, 2023; Surjanovic & Bingham, 2013), holo-bench (Stanton et al., 2024), pymoo (Blank & Deb, 2020), and Off-MOO-Bench (Xue et al., 2024). Due to the nature of synthetic function, the evaluation oracle is *analytical*, meaning that the mathematical form is known. Furthermore, the offline dataset size can be arbitrarily large (*Any*); however, researchers typically use defaults such as $50\,000$ (Chen et al., 2024) for SOO or $60\,000$ for MOO (Xue et al., 2024; Yuan et al., 2025). Similarly, many synthetic functions offer the flexibility to operate over *any* number of dimensions or objectives, and conventional practice often employs specific values (also indicated in parentheses). Finally, the design space in synthetic functions is generally *continuous*.

**Single-Objective Optimization** The SOO benchmarks are designed to assess algorithm performance across various landscapes with *many local minima*, *bowl-shaped*, *plate-shaped*, *valley-shaped*, *steep ridges/drops*, and more, each posing unique challenges to optimization (Surjanovic & Bingham, 2013).

**Multi-Objective Optimization** In addition to SOO tasks, many synthetic functions have been developed for the MOO setting, where algorithms often optimize multiple conflicting objectives simultaneously. We discuss four families of synthetic functions for MOO—*DTLZ* (Deb et al., 2005), *Omnitest* (Deb & Tiwari, 2008), *VLMOP* (van Veldhuizen & Lamont, 1999), and *ZDT* (Zitzler et al., 2000)—each characterized by distinct Pareto front properties and varying levels of complexity.

### 3.1.2 Real-World System

*Real-world system* tasks encapsulate the inherent complexity of practical engineering design. The underlying mechanisms of these designs are generally well-understood, in contrast to the scientific design tasks discussed in the next subsection. While many of these tasks still utilize analytical functions for construction, unlike the purely synthetic functions, they are derived from models of real-world systems (Tanabe & Ishibuchi, 2020). Alternatively, some tasks employ simulations (Brockman et al., 2016) or surrogate models (Tanabe & Ishibuchi, 2020) as oracles. Overall, compared to the *synthetic function* category, these tasks incur higher evaluation costs and provide less transparency into the underlying mechanisms.

**Advantages** (1) *Enhanced realism*: These tasks are constructed from real engineering systems, and they incorporate real-world constraints such as physical feasibility, manufacturability, energy consumption, and cost, thereby faithfully representing practical design challenges. (2) *Heterogeneous input representation*: The design spaces often involve a mix of continuous, categorical, and permutation-based variables, mirroring the multifaceted nature of real-world problems. This variety can more effectively test offline MBO algorithms.

**Limitations** (1) *Reduced analytical transparency*: While analytical functions offer closed-form expressions, real-world system tasks often rely on simulations or surrogate models that obscure the underlying mechanisms. (2) *High evaluation cost*: For some simulation tasks, computational expenses tend to be high, which may limit the number of feasible experiments (Brockman et al., 2016). (3) *Limited adaptability for deep learning*:

Although these tasks are more realistic than pure synthetic functions, their design is often quite basic (Tanabe & Ishibuchi, 2020) and may still lack the rich, intricate patterns necessary for deep learning methods to fully demonstrate their potential, similar to the *synthetic function* tasks.

Below we introduce *real-world system* tasks. Due to the wide range of tasks, classifying them by application area proves challenging. Therefore, we propose to categorize these tasks based on their oracle type—*analytical*, *simulation*, and *surrogate*—which we believe better reflects the inherent characteristics of each task.

**Analytical**    For some real-world system tasks, the underlying mechanisms are well understood and can be readily represented by mathematical functions. These functions serve as analytical oracles. For example, the RE2-4-1 task in the *RE suite* (Tanabe & Ishibuchi, 2020) involves minimizing both the structural volume and the joint displacement of a four-bar truss. This problem is formulated based on Newtonian physics, allowing the oracle function to be derived analytically. Other analytical tasks from the RE suite—such as various truss designs (Cheng & Li, 1999; Coello Coello & Pulido, 2005; Qian et al., 2025), reinforced concrete beams (Amir & Hasegawa, 1989), pressure vessels (Kannan & Kramer, 1994), hatch covers (Amir & Hasegawa, 1989), coil compression springs (Lampinen & Zelinka, 2000), welded beams (Ray & Liew, 2002), disc brakes (Ray & Liew, 2002), speed reducers (Farhang-Mehr & Azarm, 2002), gear trains (Deb & Srinivasan, 2006), conceptual marine design (Parsons & Scott, 2004), and water resource planning (Ray et al., 2001)—fall into this category.

The *industrial suite* benchmarks (Qian et al., 2025) further enrich this category, which are formulated as constrained optimization problems. For instance, the optimal operation task targets improvements in chemical processing quality, where the objective is to enhance the alkylating product subject to 14 constraints designed to limit onboard fuel and launcher performance. The process flow sheeting problem and process synthesis problem focus on efficient process design under practical constraints. Both objective functions and constraints are expressed mathematically for these tasks. Besides, some classic combinatorial optimization problems in industrial applications—such as the multi-objective traveling salesman problem (Lust & Teghem, 2010), the multi-objective capacitated vehicle routing problem (Zajac & Huber, 2021), and the multi-objective knapsack problem (Ishibuchi et al., 2015)—have analytical formulations that allow for rapid computation. However, these problems are not typically considered traditional design problems.

**Simulation**    Sometimes, even when the underlying mechanisms of a task are well understood, their complexity prevents us from expressing the oracle in a closed analytical form. In such cases, numerical simulations are used to obtain the oracle, despite the high computational cost often involved. A typical example is *robot morphology design*. Robot morphology design tasks focus on optimizing the structural parameters of simulated robots to maximize performance on specific tasks, as exemplified by the *Ant* and *D'Kitty* Morphology tasks (Trabucco et al., 2022). In the Ant Morphology task, the objective is to optimize 60 continuous parameters—encoding limb size, orientation, and location—for a quadruped robot from OpenAI Gym (Brockman et al., 2016) so that it runs as fast as possible. In the D'Kitty Morphology task, the goal is to adjust 56 continuous parameters defining the D'Kitty robot from ROBEL (Ahn et al., 2020) to enable a pre-trained, morphology-conditioned neural network controller, optimized using Soft Actor Critic (Haarnoja et al., 2018), to navigate the robot to a fixed location. Both tasks utilize the MuJoCo simulation (Todorov et al., 2012) to run simulations for 100 time steps and average the results over 16 independent trials, thereby providing reliable yet computationally efficient performance estimates.

This category also includes tasks for *electronics design*. These tasks focus on optimizing hardware configurations and accelerator architectures to meet specific performance and efficiency objectives, typically involving selecting parameters for processing elements, memory hierarchies, and dataflow patterns, directly influencing factors such as latency (Kumar et al., 2022). The simulator oracle is used to evaluate the performance and feasibility of accelerator designs by providing latency, power, and other metrics. Other simulation tasks include radar waveform design (Hughes, 2007) for signal processing, heat exchanger (Daniels et al., 2018) and car structure design (Kohira et al., 2018) for engineering optimization, TopTrumps (Volz et al., 2019) problem for game-based optimization, and MarioGAN (Volz et al., 2019) for procedural content generation. These tasks provide valuable and diverse benchmarks for offline MBO.

**Surrogate**  Some real-world system tasks utilize surrogate models to approximate the oracle based on simulation data, thereby reducing the cost of simulations or actual experiments. For instance, several tasks within the RE suite (Tanabe & Ishibuchi, 2020)—including vehicle crashworthiness (Liao et al., 2008), car side impact design (Jain & Deb, 2014), rocket injectors (Vaidyanathan et al., 2003), and car cab design (Deb & Jain, 2014)—employ surrogate oracles. The surrogate parameters are determined using the response surface method on data sampled from simulations. As a result, these problems differ from their original counterparts, and the inherent approximations may introduce bias that could affect the reliability of performance assessments.

### 3.1.3   Scientific Design

In contrast to the engineering problems discussed earlier, *scientific design* tasks focus on addressing fundamental scientific questions. In the context of offline MBO, these tasks span domains such as biology, chemistry, and materials science, with the aim of discovering novel designs with desired properties. The offline setting is particularly relevant here, as conducting online experiments for these designs is often prohibitively expensive.

**Advantages**  (1) *Realistic challenge*: Scientific design tasks emulate real-world discovery processes—such as protein sequence design, molecular optimization, and material discovery—that are sufficiently complex to differentiate between various offline MBO methods. Successful optimization in these domains can significantly accelerate scientific discovery and benefit humankind. (2) *Deep learning benchmarking*: Scientific design problems, including those in biological (Jumper et al., 2021) and chemical (Kuenneth & Ramprasad, 2023), exhibit rich and meaningful patterns. These characteristics provide a robust platform to reveal the potential of deep learning-based offline MBO methods.

**Limitations**  (1) *Inaccurate oracles*: Unlike synthetic functions with well-defined analytical forms, scientific design tasks often rely on approximate oracles—such as surrogate models or physics-based simulations—resulting in less convincing evaluations. (2) *Limited scalability*: The size of offline datasets in these tasks is typically quite small, making it difficult to assess the scalability of some methods. Additionally, the predetermined nature of the problem settings (e.g., fixed search spaces and objective counts) limits flexibility in comprehensively benchmarking offline MBO methods.

Below, we present specific scientific design tasks across various domains—namely *biology*, *chemistry*, and *materials science*. These benchmarks offer diverse and challenging scenarios for offline MBO, reflecting the complexity of scientific problems. Given the inherent nature of scientific design, the design space is typically discrete, and evaluation oracles are implemented via *surrogate*, *simulation*, *analytical*, or *lookup table*.

**Biology**  The *biology* category encompasses a diverse set of tasks aimed at optimizing biological sequences to enhance their performance. The primary targets in this category are *protein*, *DNA*, and *RNA* designs. Together, these tasks provide a rich testbed for evaluating the efficacy and robustness of offline MBO methods in the life sciences. Below, we describe each design in detail.

- **Protein** Proteins are sequences composed of 20 standard amino acids, and optimizing these sequences can lead to improved properties with applications in antibody development (Luo et al., 2022; Chen et al., 2025) and enzyme engineering (Hua et al., 2024). While many tasks have been proposed for protein design (see, e.g., YE et al. (2025)), here we focus specifically on those that address sequence design within the framework of offline MBO.

  (1) *Green Fluorescent Protein* (GFP): Optimizes a 237-length protein sequence to increase fluorescence, using a dataset of $56,086$ variants (Sarkisyan et al., 2016). (2) *5' Untranslated Region* (UTR) : Designs a 50-nucleotide sequence to maximize gene expression, predicting ribosome load based on $280,000$ sequences (Sample et al., 2019). (3) *Adeno-Associated Virus* (AAV): Optimizes a 28-amino acid segment of the VP1 protein using $284,000$ variants, targeting improved viral viability (Ogden et al., 2019). (4) *E4B Ubiquitination Factor* (E4B): Enhances ubiquitination activity by optimizing $100,000$ mutations (Starita et al., 2013). (5) *TEM-1 $\beta$-Lactamase* (TEM): Aims to improve thermodynamic stability, leveraging $17,857$ sequences (Ren et al., 2022). (6) *Aliphatic Amide Hydrolase* (AMIE): Focuses on boosting enzyme activity, using $6,629$ variants (Wrenbeck et al., 2017). (7) *Levoglucosan Kinase* (LGK): Targets enzyme

performance improvements, based on $7,891$ mutants (Klesmith et al., 2015). (8) *Poly(A)-Binding Protein 1* (Pab1): Optimizes RNA binding efficiency with over $36,000$ sequences (Melamed et al., 2013). (9) *SUMO E2 Conjugase* (UBE2I): Focuses on protein function optimization, based on $2,000$ variants (Weile et al., 2017). (10) *Regex*: Modifies sequences to maximize bigram counts, simulating sequence editing operations (Stanton et al., 2022). (11) *Affinity maturation* (Affinity): Mutates antibody to maximize binding affinity with the antigen (Chen et al., 2025). (12) *Red Fluorescent Protein* (RFP): A multi-objective optimization problem balancing stability and solvent-accessible surface area (Stanton et al., 2022).

In general, most protein properties are evaluated using *surrogate* oracles trained on large supervised datasets due to the high cost of direct evaluation. For some properties—such as binding affinity in (11) and stability in (12)—*simulation* oracles (e.g., Rosetta (Alford et al., 2017) or FoldX (Schymkowitz et al., 2005)) can be employed, though these simulations often lack sufficient accuracy. Similarly, properties like bigram counts can be computed using *analytical* oracles; however, such tasks tend to be relatively trivial.

- **DNA** DNA design involves the optimization of sequences comprising four nucleotides (A, C, G, and T). (1) *Transcription Factor Binding 8* (TFB8) : Optimizes transcription factor binding activity in an 8-length sequence space (Barrera et al., 2016). (2) *Transcription Factor Binding 10* (TFB10): Extends the optimization to a 10-length sequence space, aiming for higher binding affinity (Barrera et al., 2016). Trabucco et al. (2022) further process the data to ensure training set given to offline MBO methods is restricted to the bottom 50%. Oracle evaluation is performed via a *lookup table* oracle, as the properties of all possible sequences have been pre-measured.

- **RNA** RNA design involves the optimization of sequences comprising four nucleobases (A, U, C, and G). The *RNA-Binding* task is a typical example. The objective is to optimize a 14-length RNA sequence to maximize binding activity with a target transcription factor. Kim et al. (2023) adopt three target transcriptions, termed RNA-Binding-A (for L14 RNA1), RNA-Binding-B (for L14 RNA2), and RNA-Binding-C (for L14 RNA3). These properties are evaluated using the ViennaRNA package as a *simulation* oracle (Lorenz et al., 2011).

In addition to the above tasks, Stanton et al. (2024) proposes *Ehrlich* functions, a new class of closed-form test functions for biophysical sequence optimization. These functions are formulated to encapsulate key geometric properties—such as non-additivity, epistasis, and discrete feasibility constraints—that are characteristic of real-world sequence design problems. Their provable solvability, low evaluation cost, and capacity to mimic realistic biophysical interactions make Ehrlich functions a valuable benchmark in offline MBO and a promising direction for future research.

**Chemistry** The *chemistry* category comprises tasks focused on small-molecule design, in contrast to the large-molecule design tasks typical in the biology category. Numerous benchmarks for small-molecule design exist (Brown et al., 2019; Huang et al., 2021); in this paper, we restrict our discussion to those commonly used in offline MBO. Some typical examples include: (1) ChEMBL: Derived from a large-scale drug property database (Gaulton et al., 2011), this task aims to maximize molecular activity by optimizing the task-specific assay score (MCHC) value. It employs a training set of $1,093$ molecules and a discrete design space of 31-length sequences over 591 categorical values. This benchmark has been incorporated in Trabucco et al. (2022), where a random forest surrogate is used as the oracle. (2) ZINC: This benchmark provides a multi-objective molecule optimization problem involving a small molecule of roughly 128 tokens. The goal is to improve *analytical* druglikeness properties such as logP (the logarithm of the octanol–water partition coefficient, reflecting hydrophobicity) and QED (the quantitative estimate of drug-likeness) (Stanton et al., 2022). (3) Molecule: As described in Zhao et al. (2022), this task tackles a two-objective molecular generation problem, aiming to optimize activities against the biological targets GSK3$\beta$ and JNK3 in a 32-dimensional continuous latent space. Candidate solutions are decoded into molecular strings using a pre-trained decoder.

**Material Science** Unlike biological tasks focusing on large molecules such as proteins, and chemical tasks centering on small molecules, the *material science* category is dedicated to designing and optimizing materials based on complex compositional and structural performance metrics. A typical offline MBO benchmark in this domain is *Superconductor*. Derived from a dataset of 21,263 superconductors with annotated critical

temperatures (Hamidieh, 2018), it has also been incorporated into design-bench (Trabucco et al., 2022). The goal is to maximize the critical temperature by optimizing an 86-dimensional continuous representation of material composition, with a random forest surrogate serving as the evaluation oracle.

Some less common tasks include:

- **Crystal structure prediction**, which aims to identify stable crystal configurations for a given chemical composition (Qi et al., 2023). These configurations are represented through features such as lattice parameters and fractional coordinates, and a *simulation* serves as the oracle to estimate formation energies.

- **Battery materials design**, where research efforts focus on discovering better active materials for lithium-ion batteries (Valladares et al., 2021).

### 3.1.4 Machine Learning Model

We have discussed three groups of tasks: *synthetic function*, *real-world system*, and *scientific design*. In addition, a widely adopted task in the ML community is the design of *machine learning models*. This process typically involves optimizing architectures, model parameters, and hyperparameters. The evaluation is generally based on the final performance metrics obtained on a test set.

**Advantages** (1) *Practical relevance*: Enhancements in model architectures, parameters, or hyperparameters can yield significant performance improvements, with applications across deep learning, reinforcement learning, and more. (2) *Rich offline datasets*: In contrast to scientific design tasks, machine learning models often have access to extensive pre-collected datasets, enabling more robust benchmarking. (3) *Oracle evaluation*: In ML tasks, oracle evaluation is performed either via *lookup table*, *surrogate*, or through *real experiment*, offering more accurate assessments compared to the approximate evaluations typical in scientific tasks.

**Limitations** (1) *Limited search space*: Although many neural architecture benchmarks provide lookup tables, the available search spaces are often constrained and overly simplistic. In contrast, real-world architectures tend to be much larger and more complex, leading to higher evaluation costs. (2) *Oracle variability*: The inherent stochasticity in training deep learning models can introduce significant noise in performance metrics, which may obscure true improvements and complicate benchmarking.

We categorize these tasks into three major subcategories—*model architecture*, *model parameter*, and *model hyperparameter*—which are described in detail below.

**Model Architecture** *Model architecture* design, often referred to as *neural architecture search*, seeks to automatically discover high-performing network architectures by exploring vast, discrete search spaces.

- Early work (Zoph & Le, 2017) focuses on single-objective optimization and evaluate the discovered architectures via *real experiment*. The goal is to identify a 32-layer convolutional neural network with residual connections that maximizes test accuracy on CIFAR-10. The search space is defined over architectural hyperparameters such as kernel sizes, selected from $\{2, 3, 4, 5, 6\}$, and activation functions including ReLU, ELU, leaky ReLU, SELU, and SiLU, resulting in a 64-dimensional discrete space with 5 categories per dimension.

- Multi-objective NAS (MO-NAS) extends the search paradigm by simultaneously optimizing multiple performance metrics of neural architectures (Lu et al., 2023). In many MO-NAS frameworks, architectures are pre-trained and their performances recorded in *lookup tables*, enabling rapid evaluation. Typical objectives include prediction error, model complexity (e.g., number of parameters), and hardware efficiency metrics (e.g., GPU latency or FLOPs). By jointly optimizing these criteria, MO-NAS aims to identify architectures that achieve an optimal balance among accuracy, computational cost, and hardware efficiency. Representative benchmarks include *NAS-Bench-201-Test* (Krizhevsky, 2009) as well as the *C-10/MOP* and *IN-1K/MOP* suites (Lu et al., 2023).

**Model Parameter**    *Model parameters* are typically optimized via loss functions and gradient-based methods. In the context of offline MBO, these parameters are treated as high-dimensional design variables, with a particular focus on agent policy parameters. This category of tasks centers on fine-tuning the neural network weights to enhance performance.

- The *Hopper* task (Trabucco et al., 2022) aims to design a feed-forward neural network controller for a 2D hopping robot in MuJoCo (Todorov et al., 2012) to maximize the expected discounted return. The design space encompasses thousands of continuous weight parameters. Crucially, in the offline setting, the algorithm only has access to a dataset of (policy parameters, return) pairs collected from prior RL experiments, rather than interacting directly with the environment. Consequently, the policy optimization becomes a purely data-driven, model-based optimization challenge rather than a traditional learning task.

- Extending this idea to multi-objective scenarios, Xue et al. (2024) introduce *MO-Swimmer* and *MO-Hopper*, where each environment presents two conflicting objectives. In MO-Swimmer, the trade-off is between forward velocity and energy efficiency, while in MO-Hopper, it is between forward velocity and jumping height. In both cases, the full set of neural network policy weights constitutes the search space, with data collected by Xu et al. (2020). The multi-objective policy search thus aims to identify new policies that can simultaneously balance these competing criteria—without any additional simulation calls.

**Model Hyperparameter**    *Model hyperparameters*, such as learning rate, weight decay, and batch size, play a crucial role in controlling the training process. Numerous benchmarks including *HPOBench* (Eggensperger et al., 2021) exist for this task, and popular methods like Bayesian optimization (Frazier, 2018) have been widely applied. The objective is to identify the optimal hyperparameter configuration that maximizes a performance metric, typically modeled as a black-box function. To the best of our knowledge, few offline MBO methods have been developed for hyperparameter optimization. This may be because hyperparameter configurations typically involve simple scalar values, and data-driven offline MBO approaches—often relying on neural networks to capture complex design patterns—may not be ideally suited for such tasks.

### 3.2   Evaluation Metric

In this subsection, we present the *evaluation metrics* that underpin our benchmarking of offline MBO methods. Although we have described oracle evaluations when discussing the tasks, it is important to elaborate on the evaluation metrics within this context. The oracle evaluation provides a per-sample usefulness score; however, a complete assessment requires evaluating a set of samples to measure novelty and diversity. In the following, we discuss the metrics for *usefulness*, *novelty*, and *diversity*. We assume that a batch $\mathcal{B}$ of $K$ candidates (e.g., $K = 128$) is used to evaluate these metrics.

#### 3.2.1   Usefulness

The primary criterion is that the candidate set should contain high-performing solutions. In offline SOO, where only one property is considered, this *usefulness* criterion is straightforward. In contrast, in offline MOO, *usefulness* is inherently coupled with *diversity*, as the quality of the solution set is evaluated based on both its proximity to the Pareto front and its distribution along that front. We discuss these two settings separately.

**Single-Objective Optimization**    For SOO tasks, *usefulness* is evaluated by considering the normalized ground-truth scores of the candidates. Specifically, we examine both the score at the 100th percentile (i.e., the best design) and the 50th percentile (i.e., the median) as suggested in Trabucco et al. (2022). The normalized score is computed as:

$$y_n = \frac{y - y_{\min}}{y_{\max} - y_{\min}},$$

where $y$ denotes the oracle evaluation score of a design, and $y_{\min}$ and $y_{\max}$ are the minimum and maximum scores in the offline dataset, respectively. Additionally, following Jain et al. (2022), the overall performance of a candidate set $\mathcal{B}$ is also quantified by its mean score:

$$\text{Mean}(\mathcal{B}) = \frac{\sum_{(\boldsymbol{x}_i, y_i) \in \mathcal{B}} y_i}{|\mathcal{B}|},$$

where $y_i$ is the oracle evaluation for $\boldsymbol{x}_i$. This metric provides an aggregate measure of the average quality of the candidate designs.

**Multi-Objective Optimization** In the MOO setting, *usefulness* is assessed by evaluating both the proximity of the candidate set $\mathcal{B}$ to the true Pareto front and its distribution along that front, a metric that naturally also captures *diversity*. Two widely used metrics are the *hypervolume* (HV) (Zitzler & Thiele, 1998) and the *inverted generational distance* (IGD) (Bosman & Thierens, 2003).

The HV metric quantifies the size of the objective space that is dominated by the candidate set $\mathcal{B}$ and bounded by a reference point $\boldsymbol{r} = (r^1, r^2, \ldots, r^m)$. The reference point is typically chosen to be worse than any observed objective value (i.e., a nadir point). Mathematically, the HV is defined as:

$$HV(\mathcal{B}) = \text{vol}\left(\bigcup_{\boldsymbol{y} \in \mathcal{B}} \prod_{i=1}^{m} [y^i, r^i]\right),$$

where $\prod_{i=1}^{m}[y^i, r^i]$ represents an $m$-dimensional hyperrectangle (or box) spanning from the coordinates of $\boldsymbol{y}$ to the reference point $\boldsymbol{r}$ along each objective, and $\text{vol}(\cdot)$ denotes the Lebesgue measure (i.e., volume) of the union of these hyperrectangles. In simple terms, a larger hypervolume indicates that the solution set is both close to the Pareto front and well-distributed across the objective space.

The IGD metric measures the average distance from points on the true Pareto front (denoted as $PF$) to the nearest solution in the candidate set $\mathcal{B}$:

$$IGD(\mathcal{B}, PF) = \frac{1}{|PF|} \sum_{\boldsymbol{y}_{pf} \in PF} \min_{\boldsymbol{y} \in \mathcal{B}} \|\boldsymbol{y}_{pf} - \boldsymbol{y}\|.$$

Since the true Pareto front is generally unknown in real-world tasks, most existing studies primarily rely on HV to evaluate MOO performance (Tanabe & Ishibuchi, 2020; Xue et al., 2024; Yuan et al., 2025).

### 3.2.2 Diversity

*Diversity* quantifies the spread or variability within the set of generated candidate designs, and maintaining higher diversity helps reduce the risk that all candidates fail due to similar underlying limitations (Jain et al., 2022; Kim et al., 2023; Kirjner et al., 2024). In offline MBO, it is crucial not only to identify high-performing designs but also to explore diverse regions of the design space, thereby capturing multiple modes of the black-box function. A common metric for measuring diversity is the average pairwise distance between all distinct candidates in the set $\mathcal{B}$, defined as:

$$\text{Diversity}(\mathcal{B}) = \frac{1}{|\mathcal{B}|(|\mathcal{B}| - 1)} \sum_{\substack{(\boldsymbol{x}_i, y_i) \in \mathcal{B}}} \sum_{\substack{(\boldsymbol{x}_j, y_j) \in \mathcal{B} \\ (\boldsymbol{x}_j, y_j) \neq (\boldsymbol{x}_i, y_i)}} \delta(\boldsymbol{x}_i, \boldsymbol{x}_j),$$

where $\delta(\boldsymbol{x}_i, \boldsymbol{x}_j)$ denotes a distance measure (e.g., the Euclidean distance for continuous designs or the Levenshtein edit distance (Haldar & Mukhopadhyay, 2011) for discrete designs). Alternatively, the median of the pairwise distances may be used instead of the mean (Kirjner et al., 2024). A higher diversity score indicates that, on average, the generated designs are more varied within the batch $\mathcal{B}$.

Another related metric is *coverage*, as adopted in Yao et al. (2025), which evaluates how well the batch of candidate designs collectively spans the search space. It is computed as:

$$\text{L1C}(\mathcal{B}) = \frac{1}{d} \sum_{k=1}^{d} \max_{i \neq j} \left| x_{ik} - x_{jk} \right|,$$

where $d$ is the number of design dimensions and $x_{ik}$ denotes the $k$-th component of the $i$-th design $\boldsymbol{x}_i$. Note that this formulation assumes a continuous representation of the design; discrete designs must first be embedded into a continuous latent space. Intuitively, a higher coverage value indicates that the designs in $\mathcal{B}$ are more widely spread across each dimension, suggesting improved diversity.

### 3.2.3 Novelty

*Novelty* measures the degree to which the newly generated candidate set $\mathcal{B}$ differs from the offline dataset $\mathcal{D}$, thus promoting exploration beyond known designs.. In offline MBO, it is important that the proposed candidates not only perform well but also explore regions of the design space that are distinct from known designs. Novelty is typically quantified by computing the mean of the minimum distances between each candidate in $\mathcal{B}$ and the closest design in $\mathcal{D}$ (Jain et al., 2022; Kim et al., 2023). Mathematically, the novelty score is defined as:

$$\text{Novelty}(\mathcal{B}) = \frac{1}{|\mathcal{B}|} \sum_{(\boldsymbol{x}_i, y_i) \in \mathcal{B}} \min_{\boldsymbol{x} \in \mathcal{D}} \delta(\boldsymbol{x}_i, \boldsymbol{x}),$$

where $\delta(\boldsymbol{x}_i, \boldsymbol{x})$ denotes the distance metric between design $\boldsymbol{x}_i$ from $\mathcal{B}$ and a design $\boldsymbol{x}$ from $\mathcal{D}$. As before, the Euclidean distance is commonly used for continuous designs, while the Levenshtein edit distance (Haldar & Mukhopadhyay, 2011) is appropriate for discrete designs. Alternatively, one may use the median of the minimum distances instead of the mean (Kirjner et al., 2024). A higher novelty score indicates that, on average, the generated designs are more dissimilar from those in $\mathcal{D}$, thereby fostering the discovery of innovative and unexplored designs.

**Overall Summary**  Most offline MBO methods (Trabucco et al., 2021; Chen et al., 2023a) primarily focus on the *usefulness* metric, as it is the fundamental measure of candidate quality. However, *diversity* and *novelty* are also important for a comprehensive evaluation. We conjecture that this focus is partly due to the relative ease of demonstrating improvements in usefulness, whereas proving enhancements across all three metrics for a proposed method is more challenging. We encourage future research to consider all metrics.

Besides, some recent work has proposed alternative metrics beyond the three discussed above. For instance, Qian et al. (2025) introduces a *stability* metric that measures an algorithm's ability to consistently surpass the performance of the offline dataset during the optimization process. Specifically, this metric evaluates not only the final design but also all intermediate samples along the optimization trajectory, thereby assessing whether these intermediate solutions can outperform the best design in the offline dataset—a critical consideration given the challenge of determining an appropriate stopping point in offline MBO.

## 4   Surrogate Modeling

Offline MBO learns a surrogate model that approximates the black-box oracle and leverages it to guide design optimization. As shown in Eq. (2), the mean squared error loss is employed to fit the surrogate model—a loss that can also be interpreted as a maximum likelihood loss when accounting for uncertainty (Chen et al., 2024). In this work, we decompose surrogate modeling into four key components: *input representation*, *model type*, *training strategy*, and *sampling strategy*.

### 4.1   Input Representation

The *input representation* refers to the type of design $\boldsymbol{x}$, which we categorize into *continuous*, *discrete*, and *mixed* representations. Since we focus on offline MBO, the ability to compute gradients of $\boldsymbol{x}$ is a vital property of the surrogate model. For discrete and mixed representations, a common strategy is to transform them into a continuous space to facilitate gradient computation.

**Continuous**   *Continuous* representations are the most common and are often used directly in their native space due to their inherent continuity. For example, in the superconductor task, the search space is an 86-dimensional continuous vector representing the elemental composition of superconductors (Hamidieh, 2018). Other examples include Ant Morphology and D'Kitty Morphology, where the task is to optimize the morphology of quadrupedal robots using 60 and 56 continuous parameters respectively to enhance locomotion, and Hopper Controller, which involves optimizing a neural network policy with 5126 continuous weights to maximize return (Brockman et al., 2016). The continuous nature of these representations naturally facilitates gradient computation. An interesting case is image optimization; for instance, DeepDream optimizes continuous images using surrogate gradients to enhance specific features (Mordvintsev et al., 2015).

Even when using continuous inputs, it is common to map them into the latent space of a generative model for optimization and manipulation, as this latent space can better capture the semantic meaning of the design. For example, classifier (or surrogate) guidance optimizes the continuous image latent space, steering samples toward a specific category in diffusion (Dhariwal & Nichol, 2021) and flow matching models (Dao et al., 2023).

**Discrete**   In many real-world applications, the design $\boldsymbol{x}$ is *discrete*. The most common representation is *categorical encoding*, which is suitable for unordered discrete variables. Typical applications include neural architecture search, biological sequence design, and molecule design. When computing surrogate gradients on categorically encoded inputs, three strategies are typically employed. First, one may remain in the raw discrete space and use discrete gradient estimators to approximate gradients (Bengio et al., 2013; Jang et al., 2017; Chen et al., 2023b). Second, the input can be mapped into a latent space where gradients are more directly accessible (Luo et al., 2018; Gómez-Bombarelli et al., 2018). Third, the design may be transformed into a continuous representation through design-specific modeling. For instance, Liu et al. (2019); Fu et al. (2022) propose continuous relaxations of discrete representations for neural architectures and chemical structures, respectively, to facilitate gradient computation.

A less common discrete representation is *permutation encoding*, which is used when the relative order of elements is crucial. This encoding is typical in problems such as the traveling salesman problem (Lust & Teghem, 2010). However, these problems generally do not rely on surrogate models since their final solutions can be evaluated efficiently.

**Mixed**   Some optimization problems involve a combination of continuous and discrete variables, leading to hybrid input representations that require domain-specific surrogate modeling. For instance, in protein property prediction, a protein is characterized by both continuous atom coordinates and discrete residue types. A practical surrogate model is to input the discrete residue types into a pre-trained language model to obtain residue embeddings, and then feed both the residue embeddings and atom coordinates into a graph neural network for the final prediction (Wang et al., 2022; Zhang et al., 2023; Chen et al., 2023c). In this scenario, gradient computation leverages strategies developed for both continuous and discrete representations.

## 4.2   Model Type

Surrogate models can be broadly classified based on their parameterization into *parametric* and *non-parametric* models. Parametric models have a fixed number of parameters determined by their architecture, whereas non-parametric models adapt their complexity based on the available data (Hastie et al., 2009). In the context of offline MBO, *neural networks* are the predominant parametric models, whereas *kernel-based models* are the typical non-parametric choices.

**Neural Networks (NN)**   *Neural networks* are a class of parametric models where the architecture (e.g., the number of layers and neurons per layer) is predetermined prior to training (LeCun et al., 2015). These models approximate the target function by optimizing a fixed set of parameters, making them well-suited for high-dimensional and complex tasks. Owing to their generalization capabilities and scalability, NN-based surrogate models have become increasingly popular in offline MBO (Trabucco et al., 2022).

Some approaches use data-agnostic neural networks—such as multi-layer perceptrons (MLPs)—as surrogate models (Trabucco et al., 2021; Yu et al., 2021a; Yuan et al., 2023; Chen et al., 2023a). While these methods demonstrate the overall effectiveness of the optimization framework, simple MLPs may struggle to accurately capture the intricacies of complex black-box functions, especially when the design itself contains rich semantic information. To better exploit domain-specific information, recent works have designed specialized architectures tailored to the data. For example, Lee et al. (2023) employs a graph neural network (GNN) designed for modeling molecular properties and guiding molecule generation, while Chen et al. (2025) uses a protein language model to extract residue embeddings that feed into a GNN for predicting antibody binding affinity, thereby steering antibody structures towards a more stable conformation. OmniPred (Song et al., 2024) employs a specialized neural network—specifically, a language model—trained on heterogeneous offline data comprising both textual and numerical modalities, and estimates prediction uncertainty by analyzing the concentration of sampled outputs.

It is worth noting that besides neural networks, other parametric models—such as linear regression, logistic regression, polynomial regression, and support vector machines (SVM)—are also useful (Hastie et al., 2009). However, in offline MBO, these alternatives are used less frequently, as the current surge in offline MBO has largely been driven by advances in deep NNs due to their superior generalization ability and scalability.

**Kernel-Based Model**   In offline MBO, the offline dataset is often limited, making non-parametric *kernel-based models* particularly attractive. Their effective complexity increases with the number of data points, enabling them to capture intricate function behaviors while providing principled uncertainty estimates (Rasmussen & Williams, 2006). Gaussian Processes are the most widely used kernel-based models, offering closed-form posterior inference and calibrated uncertainty (Frazier, 2018). Although offline MBO does not allow further queries of the black-box function, uncertainty quantification remains crucial: it helps identify regions in the design space where the surrogate's predictions are less reliable. This information can be used to guide conservative optimization strategies that avoid over-optimistic predictions in poorly sampled areas.

Chen et al. (2022b) demonstrates that the neural tangent kernel—associated with infinite-width neural networks—can be more effective than standard kernels like the radial basis function (RBF) kernel. Furthermore, Chen et al. (2023b) introduces a kernel parameterized by pre-trained biological language models for biological sequence design. In cases where pre-trained models are unavailable, deep kernel learning can be employed to learn the kernel directly from the data, with scalability achieved via inducing points (Wilson et al., 2016).

Besides kernel-based models, other non-parametric methods—such as k-nearest neighbors (kNN) and random forests—can also serve as surrogate models. However, these approaches are typically non-differentiable and do not naturally provide robust uncertainty estimates, which makes them less suitable for gradient-based optimization in offline MBO (Hastie et al., 2009).

### 4.3   Training Strategy

To enhance generalization and robustness, surrogate models often incorporate specialized *training strategies*. We categorize these into four groups: *auxiliary loss*, *data-driven adaptation*, *collaborative ensembling*, and *generative model integration*. Note that a single method can span multiple categories due to its inherent complexity. In such cases, the method may be discussed in more than one group, with emphasis placed on the aspect most relevant to that group.

**Auxiliary Loss**   *Auxiliary losses* are incorporated into surrogate models to refine the training process by encouraging specific model behaviors, as detailed below.

*Conservatism* is a widely adopted strategy in offline MBO. Inspired by similar ideas in offline RL (Yu et al., 2021b), Trabucco et al. (2021) trains the surrogate to systematically underestimate the true objective on out-of-distribution inputs by identifying potential adversarial examples via gradient ascent and penalizing the surrogate's predictions at these points, thereby enforcing a conservative estimate. In Qi et al. (2022), offline MBO is reframed as a domain adaptation problem by treating the offline dataset as the source domain and the optimized designs as the target domain. A distributional distance loss is used to ensure that the surrogate produces conservative (i.e., less overconfident) predictions when evaluated on samples far from the offline distribution. Further, Chen et al. (2022b; 2023b) propose a bidirectional learning framework that integrates knowledge from an offline dataset into a high-performing design using both forward and backward loss mappings. Here, a kernel-based model provides a closed-form solution while the backward mapping serves as a regularizer to mitigate the impact of out-of-distribution inputs. While this method technically regularizes the designs rather than the surrogate itself, it still promotes conservatism by encouraging generated designs to remain close to the offline dataset. This principle is closely related to distributionally robust optimization (Delage & Ye, 2010), which aims to avoid overconfident decisions under uncertain region.

*Smoothness* is another important strategy for improving generalization. Yu et al. (2021a) smooths the offline data with a Gaussian filter, finds the weight perturbation that maximizes the loss, and then adjusts the model parameters to maintain local smoothness with respect to both the inputs and the weights. In the same vein, Dao et al. (2024a;b) incorporate measures of model sharpness and sensitivity, respectively, to constrain the surrogate's local behavior and improve its stability.

*Ranking Consistency* is another strategy used in offline MBO. In many cases, the surrogate is tasked with selecting promising designs rather than predicting precise objective values. To support this, Tan et al. (2025) employs a listwise ranking loss that encourages the surrogate to preserve the relative ordering of candidate designs. In scenarios where pointwise labels are noisy or unreliable—such as with augmented data—Chen et al. (2023a) adopts a pairwise ranking loss, demonstrating improved robustness over traditional mean squared error losses. In essence, ranking-based losses provide discrete, relative supervision that enhances the surrogate's robustness and decision-making reliability.

*Gradient Alignment* is exemplified by Hoang et al. (2024), which demonstrates that the effectiveness of optimization is strongly correlated with how well the surrogate's gradient field aligns with the underlying gradients of the offline data. To this end, the authors propose a dedicated loss function to enforce such alignment and improve optimization quality.

Finally, auxiliary losses naturally extend to multi-objective settings. Multi-task learning techniques such as GradNorm (Chen et al., 2018) and PcGrad (Yu et al., 2020) can be used so that learning one property via a surrogate task aids in predicting another—a valuable capability in domains like biology where labeled data is often limited (Xu et al., 2022).

**Data-Driven Adaptation**    *Data-driven adaptation* generally falls into three categories:

- **Sample Reweighting** This method assigns higher weights to samples deemed more relevant. For instance, Yuan et al. (2023) leverages bi-level optimization (Chen et al., 2022a) to learn weights for generated samples, thereby mitigating noise and enhancing surrogate. Similarly, AutoFocus calculates offline sample weights as the ratio of the probability under the search model to the initial probability, effectively refining the surrogate in the most relevant design regions (Fannjiang & Listgarten, 2020).

- **Synthetic Data Generation** Synthetic data is extensively used in offline MBO. For example, Trabucco et al. (2021) employs gradient ascent to generate adversarial designs, penalizing the surrogate on these points. Moreover, Chen et al. (2023a) and Yuan et al. (2023) apply pseudo-labeling for nearby points based on surrogate predictions, filtering out noisy samples to further improve the surrogate.

- **Domain Knowledge Injection** Incorporating domain-specific knowledge can enrich the surrogate model's understanding and enhance its extrapolation capabilities. For instance, Chen et al. (2023b) leverages a pre-trained biological language model—trained on millions of biological sequences—as a feature extractor, yielding superior performance compared to models without pre-training. Furthermore, Kuba et al. (2024b) introduces Functional Graphical Models that build a data-specific graph capturing functional independence properties, thereby imposing a structural bias that benefits black-box optimization and mitigates distribution shifts.

These data-driven techniques are applicable not only to surrogate modeling but also to generative modeling, as discussed in Section 5. In addition to *auxiliary losses* and *data-driven adaptations*, training strategies also benefit from insights drawn from *peer models* and *generative models*, as described next.

**Collaborative Ensembling**    Ensemble learning techniques combine predictions from multiple models to achieve improved performance and generalization compared to individual base learners (Hansen & Salamon, 1990; Dietterich, 2000). In the context of offline MBO, recent studies have focused on developing ensemble-based surrogate models tailored to these settings. For example, Chen et al. (2023a) and Yuan et al. (2023) employ a mean ensemble of surrogates, wherein multiple models exchange valuable sample information during optimization to enhance learning—contrasting with traditional ensembles that generally interact only during aggregation. Additionally, Fu & Levine (2021) introduces an innovative approximation of the normalized maximum-likelihood (NML) distribution to construct an uncertainty-aware forward model. For each optimization point, the approach assigns multiple labels and trains separate models on each point-label pair, with the resulting ensemble estimating the conditional NML distribution to provide robust surrogate predictions that guide the design optimization process. Furthermore, Kolli (2023) tackles gradient conflicts among ensemble members by employing multiple gradient descent steps and conflict-averse gradient descent, thereby striking a balance between conservatism and optimality.

**Generative Model Integration** Surrogate models often guide the sampling process of generative models, and in turn, several studies leverage insights from generative models to further enhance surrogate modeling. For instance, Fannjiang & Listgarten (2020) employs a variational autoencoder to model both the offline data distribution and the search model distribution, using the ratio of these probabilities as an importance weight to retrain the surrogate. Similarly, Qi et al. (2022) trains a GAN discriminator to differentiate between the offline distribution and the desired design distribution, subsequently fine-tuning the surrogate to yield mediocre predictions when designs deviate significantly from the offline data. Moreover, Chen et al. (2024) derives a conditional distribution from a diffusion model, which is used to regularize the surrogate by minimizing the KL divergence between the surrogate's output and the derived distribution.

**Practical Guide on Selecting Training Strategies** We have discussed various training strategies, which typically complement rather than conflict with each other. Choosing the appropriate strategy depends primarily on the intended use-case and the operational context of the surrogate model. Below, we provide practical guidance for selecting among these strategies:

- When using the surrogate model for gradient-based optimization (e.g. gradient ascent), conservatism-based approaches such as conservative objective models (COMs) (Trabucco et al., 2021) are particularly effective. For scenarios where the surrogate's primary role is ranking and selecting promising designs, strategies emphasizing ranking consistency are crucial (Tan et al., 2025; Chen et al., 2023a). Additionally, if the surrogate exhibits sensitivity to perturbations in inputs or weights, smoothness strategies can be prioritized (Yu et al., 2021a).

- In situations where a robust generative model is available, one may (1) employ sample reweighting, adjusting the sample weights based on the probability ratio between the offline and search distributions, thus refining surrogate training; (2) identify out-of-distribution samples by leveraging a GAN discriminator (Qi et al., 2022) or by directly computing likelihood scores (Chen et al., 2024); (3) generate additional synthetic samples with pseudo-labeling to improve the surrogate.

- Domain knowledge injection is highly effective when pre-trained, domain-specific models are accessible. These can serve as feature extractors to improve extrapolation performance (Chen et al., 2023b).

- Ensemble-based predictions generally outperform individual surrogate predictions, making collaborative ensembling effective when computational resources allow (Chen et al., 2023a; Yuan et al., 2023).

## 4.4 Sampling Strategy

Once the surrogate model is trained, gradients are computed to guide the sampling process. The typical procedure is outlined in Equation (3), with the general form given by:

$$\boldsymbol{x}_{t+1} = \boldsymbol{x}_t + \eta \cdot \mathrm{OPT}\Big(\nabla_{\boldsymbol{x}} f_{\boldsymbol{\phi}}(\boldsymbol{x})\big|_{\boldsymbol{x}=\boldsymbol{x}_t}\Big), \quad \text{for } t \in [0, T-1]. \tag{8}$$

This section discusses key aspects of the sampling process: the *learning rate $\eta$*, the *number of iterations $T$*, the optimizer *OPT*, and *test-time training*.

**Learning Rate** Selecting an optimal *learning rate $\eta$* poses a significant challenge in offline MBO due to the absence of a dedicated validation set. While Beckham et al. (2024) suggests the introduction of a validation set in offline MBO, this method may require sacrificing some high-performing data. Additionally, Chen et al. (2023b) proposes training an auxiliary model to provide weak supervision signals for optimizing the learning rate, thus enhancing sampling effectiveness. Similarly, Chemingui et al. (2024) formulates offline MBO as an offline reinforcement learning problem, where a learned policy takes the current design as input and outputs the optimal learning rate; however, this approach may still be vulnerable to reward hacking due to out-of-distribution issues in the surrogate model.

**Number of Iterations** The number of optimization steps, denoted as $T$, is another vital hyperparameter. Determining the appropriate $T$ is challenging due to the absence of a ground-truth function, which raises concerns about overfitting. This hyperparameter also correlates with the learning rate: a higher learning rate might necessitate a smaller $T$ to avoid deviating from the distribution. The strategy suggested by Trabucco et al. (2021) involves using 50 steps to generate adversarial samples and regularizing the surrogate model based on these samples. Subsequently, a similar 50-step approach is employed during the sampling phase. Meanwhile, Yu et al. (2021a); Fu & Levine (2021) report that their methods remain robust even as $T$ increases. Furthermore, Damani et al. (2023) proposes training a binary classifier to distinguish offline data from design data, observing that the degree of distribution shift depends on $T$. The classifier logits serve as a proxy for distribution shift, allowing users to constrain $T$ to regions where the surrogate predictions remain reliable.

**OPT** The term $OPT$ denotes the optimizer, which can be an algorithm such as SGD, Adam, etc. (Ruder, 2016). These optimizers are typically applied to offline SOO. In offline MOO, however, multiple gradients must be managed simultaneously. A naive approach of computing a weighted sum of these gradients often results in conflicts that hamper effective optimization. Recent methods address these conflicts: the multiple gradient descent algorithm (Désidéri, 2012) finds a common descent direction by assigning nonnegative weights that minimize the norm of the combined gradient, while PCGrad (Yu et al., 2020) resolves conflicts by projecting one gradient onto the orthogonal space of another when their inner product is negative, thereby enhancing robustness in multi-objective settings.

**Test-Time Training** A common sampling strategy involves adapting the surrogate model at the current optimization point. While this concept overlaps with the training strategies discussed in Section 4.3, the focus here is on fine-tuning the surrogate locally. Given the impracticality of optimizing the surrogate globally, it is more feasible to refine its performance near the current point. For example, Fu & Levine (2021) estimates the conditional normalized maximum likelihood by incorporating the current point into the surrogate modeling process, and (Yu et al., 2021a) adjusts the surrogate to increase local smoothness. Moreover, Chen et al. (2023a); Yuan et al. (2023) generate pseudo pairwise and pointwise labels in the neighborhood to further refine the surrogate's local behavior.

## 5 Generative Modeling

In addition to *surrogate modeling* described in Section 4, *generative modeling* plays a pivotal role in offline MBO. The high-dimensionality of design spaces renders exploration challenging, and generative models offer an effective means to navigate these spaces. As detailed in Eq. (7), offline black-box optimization can be framed as a conditional generation problem, where the objective is to model the distribution $p(\boldsymbol{x} \mid \boldsymbol{y}_c)$ with $\boldsymbol{y}_c$ representing the target conditions. By Bayes' rule, this distribution is proportional to the product of the prior $p(\boldsymbol{x})$ and the likelihood $p(\boldsymbol{y}_c \mid \boldsymbol{x})$. Broadly, two categories of conditional generation emerge in this context:

- **Inverse** This approach directly trains a conditional generative model to learn the mapping from target conditions $\boldsymbol{y}_c$ to designs $\boldsymbol{x}$, thereby capturing $p(\boldsymbol{x} \mid \boldsymbol{y}_c)$ and enabling conditional sampling.

- **Forward** This approach leverages a surrogate model for $p(\boldsymbol{y}_c \mid \boldsymbol{x})$ to steer an unconditional generative model $p(\boldsymbol{x})$ toward desirable designs. A notable special case is the use of direct gradient ascent, which bypasses the need to explicitly model the generative component $p(\boldsymbol{x})$.

In the remainder of this section, we first outline the *general principles* of these generative models—including *variational autoencoder* (VAE) (Kingma & Welling, 2014), *generative adversarial network* (GAN) (Goodfellow et al., 2014), *autoregressive model* (Vaswani et al., 2017), *diffusion model* (Ho et al., 2020), *flow matching* (Lipman et al., 2023) and *energy-based model* (EBM) (LeCun et al., 2006)—followed by a discussion on how they achieve *conditional generation* within offline MBO. Finally, we introduce *Generative Flow Network* (GFlowNet) (Bengio et al., 2023), a versatile control strategy applicable to a wide range of generative models.

### 5.1 Variational Autoencoder (VAE)

**General Principle** *Variational Autoencoders* (VAEs) integrate ideas from variational inference and autoencoders to learn a probabilistic latent representation $\boldsymbol{z}$ for the data $\boldsymbol{x}$ (Kingma & Welling, 2014). The model expresses the data likelihood as

$$p(\boldsymbol{x}) = \int p(\boldsymbol{x}|\boldsymbol{z})\, p(\boldsymbol{z})\, d\boldsymbol{z},$$

and introduces an approximate posterior $q_{\boldsymbol{\psi}}(\boldsymbol{z}|\boldsymbol{x})$ to facilitate efficient inference. In particular, the VAE adopts an encoder–decoder architecture: the encoder approximates the posterior $q_{\boldsymbol{\psi}}(\boldsymbol{z}|\boldsymbol{x})$, while the decoder models the likelihood $p_{\boldsymbol{\theta}}(\boldsymbol{x}|\boldsymbol{z})$.

**Conditional Generation** In the context of VAEs, conditional generation can be achieved through two categories of methods: inverse and forward methods. The inverse method directly trains a label-conditioned VAE to learn the mapping from target conditions to designs. For example, Brookes et al. (2019) adaptively trains the VAE on high-performing designs (such as protein sequences), enabling the direct sampling of promising candidates without the need for an external surrogate model.

In contrast, forward methods are more commonly employed, leveraging the continuous latent space of the VAE. Here, an unconditional VAE is first trained to embed designs into a continuous latent space, after which a surrogate model supplies gradient information to steer the latent codes toward regions corresponding to improved designs. This approach is particularly beneficial for discrete design optimization, as demonstrated by works on general discrete designs (Yu et al., 2021a) and discrete molecules (Gómez-Bombarelli et al., 2018), which map these designs into a latent space amenable to gradient-based optimization. Together, these methods illustrate how VAEs can effectively navigate high-dimensional design spaces, either by directly conditioning on target attributes via the inverse method or by leveraging gradient-driven manipulations within the latent space via the forward method.

We also briefly compare VAEs and normalizing flows (Kobyzev et al., 2020), as both models map designs to latent spaces and back. While VAEs rely on a learned encoder–decoder architecture, normalizing flows use carefully designed invertible operators to establish a one-to-one correspondence between the latent and input spaces. Lee et al. (2025) observes that this invertibility effectively mitigates the reconstruction gap often seen in VAEs—which can cause property discrepancies between original and reconstructed designs. Consequently, they propose a normalizing flow model for MBO, including the SeqFlow variant for sequence designs, to address these issues directly. Although the application of normalizing flows in offline MBO remains relatively limited, they represent a promising direction for future research.

### 5.2 Generative Adversarial Network (GAN)

**General Principle** *Generative Adversarial Networks* (GANs) introduce an adversarial training framework in which a generator network $G_{\boldsymbol{\theta}}(\boldsymbol{z})$ and a discriminator network $D_{\boldsymbol{\psi}}(\boldsymbol{x})$ compete against each other (Goodfellow et al., 2014). The generator maps noise $\boldsymbol{z}$ (sampled from a simple distribution, such as $\mathcal{N}(\boldsymbol{0}, \boldsymbol{I})$) into the data space, while the discriminator attempts to distinguish real data from generated samples. The standard training loss is formulated as a minimax game:

$$\min_{\boldsymbol{\theta}} \max_{\boldsymbol{\psi}} \; \mathbb{E}_{\boldsymbol{x} \sim p_{\text{data}}(\boldsymbol{x})} \left[\log D_{\boldsymbol{\psi}}(\boldsymbol{x})\right] + \mathbb{E}_{\boldsymbol{z} \sim p(\boldsymbol{z})} \left[\log\bigl(1 - D_{\boldsymbol{\psi}}(G_{\boldsymbol{\theta}}(\boldsymbol{z}))\bigr)\right].$$

**Conditional Generation** GAN-based conditional generation is typically achieved via inverse methods, owing to the absence of an inherent latent code in standard GANs. In such frameworks, both the generator and discriminator are conditioned on target labels to steer the sampling process. For instance, Kumar & Levine (2020) employs a conditional GAN where the discriminator, parameterized as $D_{\boldsymbol{\psi}}(\boldsymbol{x} \mid y)$, is trained to output 1 for valid $(\boldsymbol{x}, y)$ pairs (i.e., when $\boldsymbol{x}$ comes from the data and $y = f(\boldsymbol{x})$) and 0 for generated pairs $\bigl(G_{\boldsymbol{\theta}}(\boldsymbol{z}, y), y\bigr)$. Here, the generator acts as the inverse mapping $G_{\boldsymbol{\theta}}(\boldsymbol{z}, y)$, taking both the latent noise $\boldsymbol{z}$ and

the condition label $y$ as inputs. This setup is optimized using the following objective:

$$\min_{\boldsymbol{\theta}} \max_{\boldsymbol{\psi}} \;\; \mathcal{L}_p(\mathcal{D}) \;=\; \mathbb{E}_{y\sim p(y)}\left[\mathbb{E}_{\boldsymbol{x}\sim p_{\mathcal{D}}(\boldsymbol{x}|y)}\Big[\log D_{\boldsymbol{\psi}}(\boldsymbol{x}\mid y)\Big] + \mathbb{E}_{\boldsymbol{z}\sim p_0(\boldsymbol{z})}\Big[\log\Big(1 - D_{\boldsymbol{\psi}}\big(G_{\boldsymbol{\theta}}(\boldsymbol{z},y)\mid y\big)\Big)\Big]\right]. \quad (9)$$

This formulation corresponds to matching the true conditional distribution $p_{\mathcal{D}}(\boldsymbol{x}\mid y)$ with the model distribution $p_{G_{\boldsymbol{\theta}}}(\boldsymbol{x}\mid y)$ (obtained by marginalizing over $\boldsymbol{z}$). During guided sampling, for a given target $y$, the latent variable $\boldsymbol{z}$ is first sampled and then optimized such that the output of the forward model, $f_{\boldsymbol{\phi}}\big(G_{\boldsymbol{\theta}}(\boldsymbol{z},y)\big)$, closely approximates $y$. This process quantifies the agreement between the learned inverse map and an independently trained forward model $f_{\boldsymbol{\phi}}$, ensuring that the generated sample $G_{\boldsymbol{\theta}}(\boldsymbol{z},y)$ not only satisfies the desired condition but also lies on the valid data manifold. Importance reweighting is also employed to construct a $p(y)$ that assigns high probability to high $y$ values.

Besides these inverse methods, the discriminator of GANs is often utilized to detect whether a design is out-of-distribution (OOD), thereby regulating the optimization process of surrogate models (Qi et al., 2022; Yao et al., 2024a). Although both surrogate models and generative models have been explored in this context, these methods are technically not GAN-based forward methods, as they do not optimize the latent code within the GAN framework but rather use the discriminator to regularize the behavior of surrogate models.

### 5.3 Autoregressive Model

**General Principle** *Autoregressive models* are widely adopted for generative tasks, particularly in language modeling. Notable examples include LSTMs (Hochreiter, 1997) and Transformer-based models (Vaswani et al., 2017; Brown et al., 2020), which factorize the joint distribution in an autoregressive manner.

**Conditional Generation** Latent representations for the design exist within autoregressive models, enabling forward methods that manipulate these latents via gradient optimization. For example, Dathathri et al. (2020) proposes using gradients from a surrogate model to adjust the language model's hidden activations, thereby guiding the generation process. However, directly manipulating these latent representations is not widely adopted, likely because design properties depend on the sequence as a whole; modifying a token's latent representation without accounting for subsequent tokens can be less robust.

In autoregressive models, inverse methods are commonly applied and can be categorized into two types. The first type models *a single sequence design*, a strategy often employed in biological sequence design. For instance, Angermüller et al. (2020) utilizes an autoregressive model for biological sequences, using a surrogate as a reward and applying reinforcement learning to generate high-performing sequences. Similarly, Kim et al. (2023) employs an LSTM-based autoregressive model to generate biological sequences, then re-trains the generator using synthetic data labeled by the surrogate, assigning higher sample weights to high-performing synthetic sequences during training.

The second type models *a sequence of designs and labels*, aiming to capture the relationship between designs. In this line, Nguyen et al. (2023) pre-trains an autoregressive transformer on related and synthetic datasets and performs in-context learning by providing the offline dataset as context. A high-score label $y_c$ is then used as a query to guide design generation. Additionally, Mashkaria et al. (2023) constructs a trajectory dataset by sorting samples based on score and trains an autoregressive model on this trajectory. During sampling, the model generates candidate points by rolling out a trajectory that implicitly serves as the condition $y_c$, thereby guiding the generation process in an inverse manner.

### 5.4 Diffusion Model

**General Principle** *Diffusion models*, a subset of latent variable models, gradually perturb data by injecting Gaussian noise during the forward process. The reverse process iteratively denoises the data using a learned score estimator (Ho et al., 2020). The score function is approximated using a time-dependent neural network $s_{\boldsymbol{\theta}}(\boldsymbol{x}_t, t)$, which facilitates the transformation of noise back into samples. They have demonstrated exceptional performance in synthesizing high-fidelity images and a wide range of complex data types.

**Conditional Generation**   Conditional generation in diffusion models has been extensively studied and can also be broadly classified into two categories: inverse methods and forward methods.

In inverse methods, the conditional diffusion model is trained using either *vanilla guidance*, where the label or condition is directly provided as an input (Chang et al., 2025), or *classifier-free guidance*, which derives guidance by contrasting the outputs of a conditional model with those of an unconditional model (Ho & Salimans, 2022). For example, Zhang et al. (2024) proposes a vanilla guidance approach that learns a weight function to assign higher weights to high-performing designs. This method focuses the training of the diffusion model $s_\theta(\boldsymbol{x}_t, y_c)$ on high-performing designs, using them as implicit high-performing conditions $y_c$ during sampling. However, vanilla guidance lacks an adjustable parameter to control sampling strength, which motivates the use of classifier-free guidance where the strength can be tuned via a parameter $\omega$. The corresponding score function is defined as follows:

$$\tilde{\boldsymbol{s}}_\theta(\boldsymbol{x}_t, y_c, \omega) = (1 + \omega)\boldsymbol{s}_\theta(\boldsymbol{x}_t, y_c) - \omega \boldsymbol{s}_\theta(\boldsymbol{x}_t). \tag{10}$$

Krishnamoorthy et al. (2023) successfully apply classifier-free guidance to offline MBO by inputting the maximum value $y_c$ from the offline dataset to produce high-performing designs. Building on this, Chen et al. (2024) explore using a surrogate model to guide the parameter $\omega$. Similarly, Yun et al. (2024) extend the method by incorporating not only the target property $y_c$ but also the entire trajectory into the conditional model to steer generation. In another work, Dao et al. (2025) generate synthetic data and train diffusion models to map low-performance samples to high-performing designs; during sampling, the offline samples serve as initial samples and the diffusion model progressively guides them toward higher-performance designs.

In contrast, forward methods such as *classifier guidance* (Dhariwal & Nichol, 2021) employ a surrogate model to steer the sampling process. The score function for classifier guidance is given by:

$$\tilde{\boldsymbol{s}}_\theta(\boldsymbol{x}_t, y_c, \omega) = \boldsymbol{s}_\theta(\boldsymbol{x}_t) + \omega \nabla_{\boldsymbol{x}_t} \log p_\phi(y_c | \boldsymbol{x}_t). \tag{11}$$

In this context, Lee et al. (2023) investigate guided molecule generation toward high-performing regions with respect to target properties such as protein-ligand interactions, drug-likeness, and synthesizability. In Yuan et al. (2024), gradient ascent is first employed to optimize the design. To address potential out-of-distribution issues, the method subsequently recovers the corresponding latent representation by injecting diffusion noise and then applying a denoising procedure, yielding a sample that conforms to the diffusion prior. This approach can be interpreted as a variant of classifier guidance due to its use of classifier gradient. Compared to classifier-free guidance, classifier guidance is less frequently adopted, likely due to the additional training cost of an extra surrogate model and the potential risk of reward hacking associated with the classifier.

## 5.5   Flow Matching

**General Principle**   *Flow matching* learns a vector field $v(\boldsymbol{x}, t)$ that defines a deterministic flow by solving an ordinary differential equation (Lipman et al., 2023; Le et al., 2023). Notably, $v(\boldsymbol{x}, t)$ can be used to derive the score $\nabla_{\boldsymbol{x}} \log p_t(\boldsymbol{x})$ and vice versa (see Lemma 1 in Zheng et al. (2023)). This demonstrates that diffusion models and flow matching follow the same probability path under certain constraints.

**Conditional Generation**   Because flow matching closely resembles diffusion models, analogous conditional generation techniques can be applied. In particular, both inverse method classifier-free guidance (Zheng et al., 2023) and forward method classifier guidance (Dao et al., 2023) are readily adaptable within the flow matching framework. Regarding the inverse method, Stärk et al. (2024) introduce Dirichlet flow matching on the simplex and extend classifier-free guidance to more effectively steer the sequence generation process. In the forward method, Yuan et al. (2025) investigate the use of multiple surrogate models to guide flow sampling toward the Pareto-front in multi-objective optimization settings, with empirical results indicating superior performance over diffusion models (Yao et al., 2024b). Moreover, Chen et al. (2025) propose training an affinity predictor to steer protein conformation sampling toward stable configurations within the AlphaFlow framework (Jing et al., 2024).

Given flow matching's emerging success and its demonstrated advantages in performance and efficiency over diffusion models, we anticipate a surge in research exploring its applications to offline MBO.

### 5.6 Energy-Based Model

**General Principle** *Energy-based Models* (EBMs) define an unnormalized probability distribution over data via an energy function $E_\theta(\boldsymbol{x})$ (LeCun et al., 2006):

$$p_\theta(\boldsymbol{x}) = \frac{\exp\left(-E_\theta(\boldsymbol{x})\right)}{Z(\theta)},$$

where $Z(\theta)$ denotes the partition function. Training EBMs typically aims to assign lower energy to observed data while raising the energy of samples drawn from the model. A common objective is contrastive divergence which encourages the model to distinguish between real data and generated (negative) samples. Sampling from an EBM is challenging due to the intractability of the partition function, and it generally relies on Markov Chain Monte Carlo (MCMC) techniques (Neal, 1993; Hinton, 2002; Tieleman, 2008).

We note that GFlowNets (Bengio et al., 2023) can amortize the MCMC process, making them applicable to EBMs (Zhang et al., 2022). Although these methods enable faster mode mixing to estimate the partition function to compare with MCMC methods, thus improving the practicality of EBMs, none of these techniques have been applied to offline black-box optimization. We believe that exploring this direction could offer promising avenues for future research.

**Conditional Generation** Conditional generation in the context of EBMs can similarly be classified into inverse and forward methods. For instance, Frey et al. (2024) maps protein sequences into a latent space during an initial *jump* step and trains an EBM on this latent representation—assigning lower energy to observed data while penalizing generated samples with higher energy. In the subsequent *walk* step, Langevin MCMC is employed to sample new latent codes, with a binary projection matrix ensuring that specified regions of the sequence remain unchanged.

Similarly, Yu et al. (2024) introduces a method that jointly embeds design and properties into a compact yet expressive energy-based latent space. In this approach, the highest offline dataset score, $y_c$, is used to sample a latent code $\boldsymbol{z}$, which is then decoded to yield the design $\boldsymbol{x}$. We categorize this method as forward, since the sampling of $\boldsymbol{z}$ is governed by

$$p(\boldsymbol{z} \mid y_c) \propto p_{\boldsymbol{\theta}}(\boldsymbol{z}) \, p_{\boldsymbol{\phi}}(y_c \mid \boldsymbol{z}),$$

with an explicitly modeled surrogate whose gradient is leveraged via SVGD (Liu & Wang, 2016).

It is important to note that the distinction between forward and inverse methods in EBMs is often subtle, as both the energy function and a surrogate model essentially map a design to a scalar value. In Beckham & Pal (2023), the authors reinterpret the original forward method, conservative objective models (COMs), as an EBM trained via contrastive divergence. COMs optimizes two losses—a mean-squared error loss for surrogate modeling and a conservative objective for the EBM—using a shared network architecture, where the resulting energy function steers the sampling process towards high-performing designs, a characteristic typically associated with inverse methods. Moreover, Beckham & Pal (2023) further proposes a decoupled version of COMs, in which separate networks are employed for the surrogate and the EBM, reinforcing its classification as a forward method.

### 5.7 Control by Generative Flow Network (GFlowNet)

Unlike earlier sections on general generative models and their conditional approaches, this section introduces a *control* method: *Generative Flow Networks* (GFlowNets) (Bengio et al., 2021; 2023). GFlowNets model generation as a sequential decision process, where a solution $\boldsymbol{x}$ is constructed through a trajectory of transitions. Their goal is amortized inference—sampling from a distribution proportional to a reward function, $p(\boldsymbol{x}) \propto R(\boldsymbol{x})$. This produces samples that are both high-reward and diverse, which is especially important when the reward is an imperfect surrogate learned from offline data. Unlike standard RL, which focuses on maximizing $R(\boldsymbol{x})$ and may be unsafe under epistemic uncertainty, GFlowNets balance exploration and exploitation, making them attractive in offline black-box optimization. A sample is generated by traversing a directed acyclic graph from an initial state $\boldsymbol{s}_0$ to a terminal state $\boldsymbol{s}_T = \boldsymbol{x}$:

$$\boldsymbol{s}_0 \to \boldsymbol{s}_1 \to \cdots \to \boldsymbol{s}_T.$$

Training is based on the *flow consistency* condition: for any intermediate state $\boldsymbol{s}'$, incoming flow equals outgoing flow,

$$\sum_{\boldsymbol{s}\in\text{Parents}(\boldsymbol{s}')} F(\boldsymbol{s};\theta)\,P_F(\boldsymbol{s}'\mid\boldsymbol{s};\theta) = F(\boldsymbol{s}';\theta),$$

with terminal flow fixed to the reward,

$$F(\boldsymbol{s}_T;\theta) = R(\boldsymbol{x}).$$

Different training objectives instantiate this principle, including trajectory balance (TB) (Malkin et al., 2022) and sub-trajectory balance (SubTB) (Madan et al., 2023). These objectives have been used to control diverse generative models, including molecular graphs (Bengio et al., 2021), bidirectional string models (Kim et al., 2024c), and diffusion models (Lahlou et al., 2023a).

**Offline MBO applications**   During sampling, a GFlowNet sequentially selects transitions based on its learned policy until reaching a terminal state $\boldsymbol{x}$. This sequential framework is particularly effective for exploring high-dimensional spaces and generating diverse candidate solutions. In offline MBO, guided sampling via GFlowNets is especially natural.

For example, Jain et al. (2022) generates desirable biological sequences from scratch by defining a target reward function as the upper confidence bound score from a surrogate model. To improve robustness against the imperfections of the surrogate model at early stages, Kim et al. (2024a) conservatively search promising regions by introducing a parameter $\delta$, which is adjusted based on the prediction model's uncertainty. Additionally, Ghari et al. (2023) utilize GFlowNets to modify existing sequences to enhance target properties, while Jain et al. (2023) employ conditional GFlowNets to generate diverse Pareto-optimal solutions for multi-objective optimization problems. Note that they use autoregressive models to produce these biological sequences and employ GFlowNets to control them effectively.

One of the main challenges in offline black-box optimization is the exploration-exploitation trade-off. Unlike RL, GFlowNets can effectively balance exploration and exploitation using a temperature parameter $\beta$, i.e., $p(\boldsymbol{x}|\beta) \propto R(\boldsymbol{x})^\beta$. Kim et al. (2024b) propose an effective way to learn such a policy by introducing a logit scaling network and verify that it achieves high extrapolation ability in offline black-box optimization tasks. They use directional token generative models, which can generate sequences by prepending or appending tokens, and employ GFlowNets to control them.

We can also formulate the offline black-box optimization problem as posterior inference, where we have a pre-trained policy $p(\boldsymbol{x})$ and a reward function $R(\boldsymbol{x})$ trained on a given dataset. In this setting, we can use the RTB loss to obtain an unbiased sampler that amortizes the posterior distribution. For example, Yun et al. (2025) propose a posterior inference method with GFlowNets for diffusion models in black-box online black-box optimization. This work may provide a key insight for applying posterior inference with GFlowNets to offline black-box optimization.

**Comparison of Generative Models in Offline MBO.**   We have presented a range of generative models used in offline MBO. Below, we summarize their respective strengths and limitations in this context.

- Variational autoencoders provide a continuous, differentiable latent space suitable for gradient-based offline MBO, but they can suffer from posterior collapse and reconstruction mismatch that may lead to inaccurate property estimates.

- Generative adversarial networks generate high-quality samples but typically lack an explicit latent space for controllable optimization, often requiring inverse methods. Additionally, training can be unstable due to mode collapse. The GAN discriminator is frequently used to detect out-of-distribution (OOD) designs.

- Autoregressive models are well-suited for sequence generation and are often combined with reinforcement learning techniques for guided sampling, though the sequential sampling process is inherently slow.

- Diffusion models provide stable training, high sample fidelity, and natural diversity. They are computationally intensive at inference time due to their iterative denoising process, but both classifier and classifier-free guidance strategies have been effectively adapted to offline MBO.

- Flow matching shares many benefits with diffusion models but offers improved efficiency and performance.

- EBMs are uniquely valuable in offline MBO for their ability to naturally penalize out-of-distribution samples and flexibly encode complex, domain-specific objectives. However, they are limited by slow MCMC-based sampling and unstable training dynamics.

- GFlowNets, as a control framework, are particularly well-suited for offline MBO due to their ability to generate diverse, high-reward solutions and explore safely under uncertainty, though training cost is relatively high due to flow consistency constraints.

**Practical Guide on Selecting Generative Models.** We have discussed various generative models. The choice of an appropriate model primarily depends on the data modality. Each type of data typically has a community-endorsed, state-of-the-art generative model. For instance, when the design is a prompt and the black-box property is the downstream task performance of an LLM given that prompt, autoregressive GPT-style models are the dominant choice due to strong empirical performances (Yang et al., 2024). Similarly, when the design is an image and the target property is a fine-grained visual attribute (e.g., a specific hairstyle), diffusion models are preferred for their superior image generation capabilities (Dhariwal & Nichol, 2021).

While model selection is primarily guided by data modality, this does not diminish the importance of discussing conditional generation strategies across different model families. In offline MBO, each generative model typically requires a tailored conditioning approach. For example, RGD (Chen et al., 2024) exploits the probability flow ODE of diffusion models to derive a refined proxy distribution and optimize the strength parameter—an approach unique to diffusion models. Similarly, MIN (Kumar & Levine, 2020) utilizes a GAN to directly map high-scoring property values to candidate designs through one-step generation, optimizing the values via a forward-inverse consistency loss, which is incompatible with the multi-step sampling process of diffusion models. Furthermore, certain models like VAEs support latent code manipulation, a feature generally unavailable in standard GANs due to the lack of an explicit latent representation.

There are cases where multiple generative models perform well for the same data modality. Protein sequence design is a representative example, and it is one of the most extensively studied tasks in offline MBO. In this setting, both autoregressive (AR) models (Elnaggar et al., 2021) and diffusion models (Wang et al., 2024) have demonstrated strong empirical performance. When deciding between the two, several additional considerations become important:

- **Likelihood estimation:** Offline MBO often requires estimating the likelihood of candidate designs to detect and avoid out-of-distribution (OOD) samples. AR models can compute exact likelihoods by factorizing the joint distribution into conditional probabilities. In contrast, diffusion models rely on variational approximations such as the ELBO, which is computationally challenging. Thus, when precise likelihood computation and rigorous OOD detection are essential, AR models are generally preferred.

- **Sample diversity:** Generating diverse samples is crucial for effective exploration in offline MBO. Diffusion models naturally provide broader mode coverage compared to AR models, making them better suited for tasks where diversity and exploration are prioritized.

- **Sample manipulation:** The ability to perform localized edits on generated samples is critically important for control in offline MBO. Diffusion models support such fine-grained manipulation through their iterative denoising process. For instance, in antibody design, diffusion models can selectively generate CDR regions while preserving the antibody framework (Luo et al., 2022)—a capability not easily realized with AR models. Moreover, diffusion models benefit from their bidirectional nature (conditioning on both past and future context) and are compatible with advanced guided sampling techniques, enabling more flexible and nuanced control. A trade-off, however, is that diffusion models often have a fixed sequence length, whereas AR models natively support variable-length generation.

- **Availability of pretrained models:** Practical deployment is often influenced by the accessibility of strong pretrained models. Fortunately, both AR (Elnaggar et al., 2021) and diffusion (Wang et al., 2024) models are publicly available and widely used. Leveraging these publicly available models can help mitigate OOD issues effectively.

- **Sampling efficiency:** Efficient sampling is essential for scaling offline MBO to large candidate sets. AR models generate sequences token by token and are generally faster for shorter sequences. Diffusion models denoise entire sequences over multiple steps, which can be more efficient for longer sequences, though their overall speed depends on the number of denoising iterations used.

**Summary:** In offline MBO for protein sequence design, diffusion models may be more suitable due to their superior sample diversity and manipulability, despite limitations in exact likelihood estimation.

# 6 Discussion and Future Direction

In this paper, we present a comprehensive review of offline MBO. Despite significant efforts to develop robust surrogate models and generative models, many practical challenges remain. Offline MBO is inherently difficult due to the limited availability of offline datasets and the epistemic uncertainty that plagues surrogate models. In this section, we outline promising future research directions aimed at addressing these challenges, including the development of more rigorous benchmarks, improved uncertainty estimation methods, innovative graphical modeling approaches for surrogate modeling, advanced generative modeling techniques, and high-impact applications in LLM alignment and AI safety.

**Robust and Realistic Benchmarking on Scientific Design** Current *scientific design* benchmarks in offline MBO face two major challenges. First, due to the difficulty of collecting labeled data, some benchmarks—such as TFB8 and TFB10 (Barrera et al., 2016)—offer overly constrained search spaces where even simple gradient ascent methods can achieve impressive results, making it difficult to distinguish the performance of more sophisticated algorithms. Second, benchmarks like superconductor (Hamidieh, 2018) often rely on learned oracles for evaluation, which can be vulnerable to manipulation and may not accurately reflect true performance. Moving forward, it is essential to develop benchmarks that not only present more challenging search spaces but also incorporate reliable evaluation protocols that are resistant to exploitation.

**Uncertainty Estimation of Surrogate Model** In offline black-box optimization, capturing the significant epistemic uncertainty in surrogate models is paramount, especially because identifying which inputs drive this uncertainty remains a core challenge. While uncertainty estimation is similarly important in online black-box optimization, in offline scenarios it is even more critical for preventing reward hacking, as one cannot correct for flawed model predictions through active data collection. Various techniques—such as adversarial regularization (Trabucco et al., 2021), smoothness priors (Yu et al., 2021a), and kernel-based methods (Chen et al., 2022b)—have been proposed to mitigate uncertainty and ensure safer optimization; however, there has been comparatively little focus on leveraging Bayesian methods for directly quantifying epistemic uncertainty. Moreover, existing benchmarks often emphasize overall optimization performance without clarifying whether observed gains stem from superior surrogate modeling, improved optimization strategies, or mere chance. This lack of distinction underscores the need for independent and rigorous evaluations of the uncertainty estimation capabilities of the surrogate model in newly developed algorithms. Although recent efforts—such as Jain et al. (2022) using Monte Carlo (MC) dropout (Gal & Ghahramani, 2016) and deep ensembles (Lakshminarayanan et al., 2017), and Kim et al. (2024a) incorporating uncertainty measures for conservative search—have made inroads into this area, they still fall short of providing a robust solution. Consequently, advancing offline MBO demands a fundamentally stronger approach to uncertainty quantification that transcends basic MC dropout or ensembling techniques.

Future work may build upon this line of research by exploring recent methodologies such as:

- **DEUP: Direct Epistemic Uncertainty Prediction** (Lahlou et al., 2023b): This approach directly estimates the excess risk associated with model misspecification by learning a secondary predictor for generalization error, offering a more principled uncertainty measure.

- **Efficient Variational Inference Methods over Neural Network Parameters**: Efficient variational inference methods offer promising solutions for tackling the intractable posterior inference in neural networks. For example, the Variational Bayesian Last Layer (VBLL) approach (Harrison et al., 2024)

performs inference solely on the final layer, reducing the computational complexity to a quadratic level. Similarly, GFlowOut, introduced by Liu et al. (2023), leverages GFlowNets—a form of hierarchical variational inference (Malkin et al., 2023)—to approximate the posterior distribution over dropout masks, effectively addressing challenges related to multi-modality and sample dependency. Together, these methods represent compelling candidates for enhancing uncertainty estimation in surrogate models while maintaining computational efficiency.

- **Deep Kernels**: Deep kernels, which leverage deep architectures to learn expressive kernel functions, offer a scalable alternative to Gaussian processes (Wilson et al., 2016; Liu et al., 2020). They have been applied to biological sequence design scenarios by employing a denoising autoencoder to learn a discriminative deep kernel GP for Bayesian optimization of biological sequences (Stanton et al., 2022). This approach was further refined by Gruver et al. (2023), who used a discrete diffusion model and simple last-layer ensemble techniques to evaluate uncertainty. These successes in online black-box optimization could be directly transferred to offline MBO, where uncertainty estimation helps prevent reward hacking. Such a strategy could effectively combine existing surrogate modeling and generative modeling methods in offline MBO fields.

**Graphical Surrogate Model**  Offline surrogate models often suffer from limited data coverage, making them vulnerable to overestimation and reward hacking in regions outside the offline distribution. A promising mitigation strategy is to leverage factorized graphical models—such as Functional Graphical Models (FGMs) (Kuba et al., 2024b)—to decompose a high-dimensional function into a sum of local subfunctions defined over smaller cliques. This structured factorization enables each subcomponent to be learned from a subspace with denser data coverage, thereby localizing distribution shifts and enhancing both robustness and OOD generalization (Kuba et al., 2024a). In addition to FGMs, candidate methods for graph discovery – such as those based on GFlowNets –offer alternative avenues to uncover latent causal structures (Deleu et al., 2022). Employing such graphical discovery methods will provide a principled means to address distributional shifts in offline MBO, ultimately leading to more reliable optimization outcomes under limited data coverage.

**Advanced Generative Modeling**  Many existing offline MBO methods employ relatively simplistic generative models. For instance, several approaches use basic encoder-decoder architectures that map protein sequences into a latent logit space for optimization (Trabucco et al., 2022). While such methods capture general machine learning principles, they often overlook the rich biophysical information embedded in the data. In applications like protein design, it is crucial to leverage models that integrate domain-specific knowledge—such as pre-trained language models or structure-aware representations—to more accurately model the underlying data distribution (Watson et al., 2023). By developing generative models that are tailored to the unique characteristics of the target domain, future methods can achieve not only improved optimization performance but also greater interpretability and robustness in real-world applications.

**Application to LLM Alignment and AI Safety**  While offline MBO has traditionally been applied to design tasks in fields such as biology or chemistry, its methodologies can also potentially inform post-training strategies for large language models (LLMs). Approaches like supervised fine-tuning (SFT), RLHF (Ouyang et al., 2022), or RL-based reasoning (Guo et al., 2025), aim to enhance text generation by leveraging reward signals derived from human preference models or other evaluative metrics. Because these reward models are inherently uncertain and prone to reward hacking, adopting the conservative and safe optimization techniques developed in offline MBO may help mitigate such risks. Looking forward, we speculate that insights from offline MBO's robust optimization principles may also inform future directions in ensuring the safety and reliability of increasingly capable LLMs (Bengio et al., 2025).

## Acknowledgement

This research is partially supported by the Fonds de recherche du Québec – Nature et technologies, as well as funding from Samsung, CIFAR, the CIFAR AI Chair program, and the Canadian AI Safety Institute Research Program at CIFAR through a Catalyst award. Minsu Kim acknowledges funding from the KAIST

Jang Young Sil Fellow Program. We sincerely thank Moksh Jain from Mila and Brandon Trabucco from CMU for insightful discussions.

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
