# OpenReview forum: "Offline Model-Based Optimization: Comprehensive Review"
_TMLR — Accepted by TMLR_

### Review · Reviewer_JsRc · 2025-10-19

**Summary Of Contributions:**

**Summary**

This paper provides a comprehensive literature review of offline model-based optimization (MBO). This field aims to optimize a black-box function based on a pre-defined dataset of input-output pairs and some offline-specific model. The paper investigates different aspects of offline MBO, including problem definition/classification, dataset (benchmark), existing approaches, and future direction.

**Audience:**

Yes

**Audience Explanation:**

**Strength**

This survey paper is timely and easy to follow.

**Broader Impact Concerns:**

N/A.

**Claims And Evidence:**

No

**Claims Explanation:**

**Weaknesses**

As a newcomer to the field of offline MBO (the target audience of the paper), after reading through the survey, I have several concerns that I would like the authors to address.

1. Inconsistency of notation/terminology

   This is one of the most significant issues of this survey. As a newcomer in the field, I expect the survey to serve as a self-contained guide through the basic problem setup and the high-level ideas behind the existing approaches. However, this survey contains several inconsistencies in terms of notation and terminology, making it sometimes confusing to read. Below, I detail some of the problems I noticed.

   - Offline MBO and offline black-box optimization

     As suggested by the title, this paper is a survey of offline model-based optimization. However, this survey mixes up offline MBO and offline black-box optimization in many places. Although in my understanding, offline MBO builds on offline black-box optimization, it would be beneficial to be precise about their relation (e.g., offline MBO is a method for offline black-box optimization). Otherwise, it looks confusing when the paper goes back and forth between these two notions.

   - Use of "surrogate model"

     The concept of surrogate model appears in two places in this paper, but they seem to carry different meanings. In the **Benchmark** section, surrogate models refer to the approximation of the true oracle, which is different from their meaning in the **Surrogate modeling** section. This inconsistency can cause much confusion when reading the paper for the first time.

   - Terminology abbreviation/Undefined terminology

     Some of the terminologies are abbreviated, but they are not properly defined when they appear for the first time. For example, out-of-distribution (OOD), upper confidence bound (UCB), GAN.

     Besides, the paper seems to introduce a number of concepts without explaining their definition. For example, "spurious optimization", "epistemic uncertainty", "aleatoric", "lower confidence bound", "many-objective", "COMs"

2. Duplicate contents

   Some of the contents of the survey are repeated many times. For example, a **Generative modeling** paragraph appears in the **Problem definition** section. These repetitions often make things unclear.

3. Inaccurate references

   I also find a number of inaccuracies in the references of the paper. Since this paper is a literature review, I strongly recommend that the authors carefully check all the references in the paper and ensure they are timely and accurate. You can use an LLM to assist in transforming the bib file.

   - Some of the journal names are not correct. For example,
     1. (Capitalization) Journal of chemical theory and computation
     2. (Capitalization) Journal of chemical information and modeling
     3. (Capitalization) PMLR
     4. (Capitalization) Operations research
     5. (Abbreviation) Proc. Adv. Neur. Inf. Proc. Syst.
     6. There are numerous issues like this. Please carefully check.
   - Some papers are already published, but the outdated arXiv version is used. For example,
     1. Posterior inference with diffusion models for high-dimensional black-box optimization
     2. On the design fundamentals of diffusion models
     3. AffinityFlow: guided flows for antibody affinity maturation
     4. Differentiable scaffolding tree for molecular optimization
   - There are several duplicated references.
     1. Survey of variation in human transcription factors reveals prevalent DNA binding changes.
     2. Human 5 utr design and variant effect prediction from a massively parallel translation assay.
     3. Local fitness landscape of the green fluorescent protein
     4. Objectives and methods in multi-objective routing problems: a survey and classification scheme

Overall, I find this survey paper, in its current form, is not sufficiently well-written and friendly for a newcomer to the field. I suggest the authors carefully address the above issues before considering publishing it.

**Questions**

1. In the **Evaluation Metric** section, only usefulness is commonly used in the literature. Although you mentioned that it is "partly due to the relative ease of demonstrating improvements in usefulness", I find the motivation behind the other two metrics is not very clear. Could you elaborate more on them?
2. The section **Sampling Strategy** mostly discusses how to optimize the surrogate model. Why is it called "sampling strategy"?
3. The **Conclusion** section (**Robust and Realistic Benchmarking**) does not follow the structure of the previous discussion. Please consider reorganizing it.

**Minor issues**

1. Page 1

   "offline black-box optimization" is not properly capitalized throughout the paper.

2. Page 1

   "beyond training data", "training distribution". The concept of "training data" is not explicitly defined.

3. Page 1

   "High-performing", "high-performance", and "high-value" are used inconsistently throughout the paper.

4. Page 3

   Sampling strategy/Number of **itertions** => Sampling strategy/Number of **iterations**.

5. Page 4

   Incorrect full stop in equation (1).

6. Page 6

   "the focus is on a single property". The meaning of "property" here is unclear.

7. Page 6

   "online black-box optimization" is not properly capitalized

8. Page 6

   The order of exploration and exploitation can be exchanged.

9. Page 8

   "can be arbitrarily large (Any)" is unclear.

10. Page 8

    "(as indicated in parentheses)" is unclear.

11. Page 8

    The definition of "DTLZ, Omnitest, VLMOP, and ZDT" is not given.

12. Page 10

    materials science => material science.

13. Page 11

    The description of the **DNA** bullet point is not clear.

14. Page 11

    I don't think highly technical terms like "CHEMBL3885882" should appear here.

15. Page 12

    Why is "leading to higher evaluation costs" discussed as "Limited search space"?

16. Page 18

    employs a variational autoencoder => employ a variational autoencoder.

17. Page 19

    Why does scaling by $\sqrt{d}$ compensate for the increase in the gradient norm? Intuitively, one should decrease the stepsize when the gradient norm becomes too large.

18. Page 20

    Is there a reference for "necessitate a smaller $T$"?

19. Page 21

    MBO is redefined

20. Page 21, 22

    The format of the boldface letters is inconsistent in the **Conditional generation** section.

**Requested Changes:**

See **Weaknesses** and **Questions/Minor issues**

---

> ### Author Response · Authors · 2025-10-31
> **Point-by-Point Response to Reviewer JsRc (1/3)**
>
> Dear Reviewer JsRc,
>
> We thank you for your thoughtful and constructive comments, which we have carefully addressed and incorporated into the revised manuscript. All changes are highlighted in red.
>
> ## Weaknesses
>
> > 1. Inconsistency of notation/terminology: This is one of the most significant issues of this survey. As a newcomer in the field, I expect the survey to serve as a self-contained guide through the basic problem setup and the high-level ideas behind the existing approaches. However, this survey contains several inconsistencies in terms of notation and terminology, making it sometimes confusing to read. Below, I detail some of the problems I noticed.
>
> We agree that consistent notation and terminology are essential for a self-contained survey. Below we clarify and revise the corresponding parts accordingly.
>
> > Offline MBO and offline black-box optimization: As suggested by the title, this paper is a survey of offline model-based optimization. However, this survey mixes up offline MBO and offline black-box optimization in many places. Although in my understanding, offline MBO builds on offline black-box optimization, it would be beneficial to be precise about their relation (e.g., offline MBO is a method for offline black-box optimization). Otherwise, it looks confusing when the paper goes back and forth between these two notions.
>
> As recommended, we now clearly define the relationship between the two concepts in the introduction. Specifically, we state that "Offline model-based optimization (MBO) is a paradigm that leverages the generalization capabilities of deep neural networks to develop offline-specific surrogate and generative models for solving offline black-box optimization problems." This definition has been explicitly introduced in the introduction of the revised manuscript, and the terminology has been updated throughout to ensure consistency and clarity.
>
> >  Use of "surrogate model": The concept of surrogate model appears in two places in this paper, but they seem to carry different meanings. In the Benchmark section, surrogate models refer to the approximation of the true oracle, which is different from their meaning in the Surrogate modeling section. This inconsistency can cause much confusion when reading the paper for the first time.
>
> Our intended meaning of “surrogate model” is indeed the approximation of the true oracle (black-box function) throughout the paper. The previous version did not make this explicit in the Surrogate Modeling section. To address this, we have added the sentence “Offline MBO learns a surrogate model that approximates the black-box oracle and leverages it to guide design optimization.” at the beginning of the section and revised the manuscript to maintain consistent terminology.
>
> > Terminology abbreviation/Undefined terminology: Some of the terminologies are abbreviated, but they are not properly defined when they appear for the first time. For example, out-of-distribution (OOD), upper confidence bound (UCB), GAN. Besides, the paper seems to introduce a number of concepts without explaining their definition. For example, "spurious optimization", "epistemic uncertainty", "aleatoric", "lower confidence bound", "many-objective", "COMs"
>
> We have ensured that all abbreviations (e.g., OOD, UCB, GAN) are defined upon first use and appropriately cited. Moreover, we have clarified previously undefined terms such as “spurious optimization,” “epistemic/aleatoric uncertainty,” “lower confidence bound,” and “COMs,” each with relevant references.
>
> > 2. Duplicate contents: Some of the contents of the survey are repeated many times. For example, a Generative modeling paragraph appears in the Problem definition section. These repetitions often make things unclear.
>
> We understand your concern about repetition. However, we believe introducing generative modeling both in the Problem Definition section and in the dedicated Generative Modeling section serves different purposes.
>
> - In the **Problem Definition** section, we introduce generative modeling conceptually—as a novel conditional-generation view of offline MBO—to contrast it with the traditional surrogate-modeling perspective.
> - In the **Generative Modeling** section, we provide a detailed discussion of algorithms and recent methodological advances.
>
> While this dual mention may appear repetitive, it is intentional—first to present a novel conditional-generation view of offline MBO and later to provide methodological depth. To clarify this structure, we have added the note: “We discuss generative modeling here only at a high level to provide a novel conditional-generation view of offline MBO, and revisit it for a detailed exposition of methodological developments.”

---

> ### Author Response · Authors · 2025-10-31
> **Point-by-Point Response to Reviewer JsRc (2/3)**
>
> > 3. Inaccurate references: I also find a number of inaccuracies in the references of the paper. Since this paper is a literature review, I strongly recommend that the authors carefully check all the references in the paper and ensure they are timely and accurate. You can use an LLM to assist in transforming the bib file.
>
> We appreciate your comment. As a literature survey, maintaining accurate and up-to-date references is essential. We have carefully reviewed and corrected all citations, ensuring proper capitalization, publication status, and consistency through both automated checks and manual validation.
>
>
>
> ## Questions
>
> > 1. In the Evaluation Metric section, only usefulness is commonly used in the literature. Although you mentioned that it is "partly due to the relative ease of demonstrating improvements in usefulness", I find the motivation behind the other two metrics is not very clear. Could you elaborate more on them?
>
> You are right that usefulness is the most widely used metric.
> However, in real discovery tasks—whether designing molecules, materials, or algorithms—we rarely want only the single best-scoring item. We usually aim to generate a set of promising candidates that are useful, diverse, and novel.
>
> (1) Usefulness means each candidate performs well for the intended goal (e.g., a protein that can effectively inhibit a bacterium); (2) Diversity means the candidates differ from each other. If all candidates are too similar, they may fail for the same reason in real experiments, so diversity increases the likelihood that at least one candidate will succeed; (3) Novelty means the designs are distinct from existing ones. In the protein case, this helps avoid rediscovering or resembling known designs that bacteria have already developed resistance to—meaning those older variants no longer work effectively.
>
> Optimization that focuses solely on usefulness can easily lead to redundancy or mode collapse, limiting real-world discovery. We have clarified this motivation in the revised Evaluation Metrics section.
>
> > 2. The section Sampling Strategy mostly discusses how to optimize the surrogate model. Why is it called "sampling strategy"?
>
> In the Sampling Strategy section, the surrogate model is kept fixed in most cases—our focus is on optimizing the design variables, as shown in Eq. (8). The discussions on learning rate, number of iterations, and optimizer (OPT) therefore pertain to the optimization of the design, rather than to the training of the surrogate model.
>
> > 3. The Conclusion section (Robust and Realistic Benchmarking) does not follow the structure of the previous discussion. Please consider reorganizing it.
>
> As the discussion on Robust and Realistic Benchmarking mainly concerns scientific design tasks, we have revised and reorganized this section for better structural consistency.

---

> ### Author Response · Authors · 2025-10-31
> **Point-by-Point Response to Reviewer JsRc (3/3)**
>
> ## Minor issues
>
> > Page 1 "offline black-box optimization" is not properly capitalized throughout the paper.
>
> We have corrected the capitalization of “offline black-box optimization” throughout the paper.
>
> > Page 1 "beyond training data", "training distribution". The concept of "training data" is not explicitly defined (TODO).
>
> The terms "training data" and "training distribution" refer to the offline dataset and its underlying distribution. We have revised them to "offline data" and "offline distribution" for greater clarity.
>
> > Page 1 "High-performing", "high-performance", and "high-value" are used inconsistently throughout the paper.
>
> We have corrected the use to use "high-performing" consistently.
>
> > Page 3 Sampling strategy/Number of itertions => Sampling strategy/Number of iterations.
>
> We have corrected the typo “itertions” to “iterations” on Page 3.
>
> > Page 4 Incorrect full stop in equation (1).
>
> We have removed the incorrect full stop in Equation (1).
>
> > Page 6 "the focus is on a single property". The meaning of "property" here is unclear.
>
> We have clarified the meaning of “property” by adding an example: in neural architecture design, the property may refer to model accuracy, while in multi-objective settings additional properties such as efficiency are considered.
>
> > Page 6 The order of exploration and exploitation can be exchanged.
>
> Although the order can be exchanged, we retain the conventional phrasing “exploration and exploitation” following standard RL terminology.
>
> > Page 8 "can be arbitrarily large (Any)" is unclear.
>
> “Any” means that the benchmark does not impose a fixed dataset size — users can generate arbitrarily large offline datasets.
>
> > Page 8 "(as indicated in parentheses)" is unclear.
>
> We have removed the phrase “(as indicated in parentheses)” to improve clarity.
>
> > Page 8 The definition of "DTLZ, Omnitest, VLMOP, and ZDT" is not given.
>
> We have added citations for DTLZ, Omnitest, VLMOP, and ZDT to clarify their origins and definitions.
>
> > Page 10 materials science => material science.
>
> We retain “materials science,” which is the standard and widely accepted term in the field.
>
> > Page 11 I don't think highly technical terms like "CHEMBL3885882" should appear here.
>
> We have removed the term.
>
> > Page 12 Why is "leading to higher evaluation costs" discussed as "Limited search space"?
>
> Because when the evaluation cost is low, it is feasible to exhaustively evaluate all candidate designs in a large search space, effectively constructing a lookup oracle.
>
> > Page 19 Why does scaling by compensate for the increase in the gradient norm? Intuitively, one should decrease the stepsize when the gradient norm becomes too large.
>
> The step-size scaling likely stems from the weight initialization with variance ($1/d$), which makes the input gradient scale as ($1/\sqrt{d}$); multiplying by ($\sqrt{d}$) thus yields (O($1$)) updates per dimension. As no formal reference was found, we have removed this sentence and contacted the original authors for clarification.
>
>
> > Is there a reference for "necessitate a smaller T"?
>
> A higher learning rate roughly mimics multiple smaller updates, so with a large (T), it can lead to overfitting to the surrogate—hence a smaller (T) is preferred.
>
> > Page 21 MBO is redefined
>
> We have corrected that.
>
> > Page 21, 22 The format of the boldface letters is inconsistent in the Conditional generation section.
>
> We have unified the formatting of all boldface symbols for consistency.
>
> ## Concluding Remark
>
> We thank you again for your thoughtful and constructive comments. We look forward to hearing your feedback during the rebuttal period.

---

> > ### Author Response · Authors · 2025-11-28
> >
> > Dear Reviewer JsRc,
> >
> > We sincerely appreciate your thoughtful comments and have updated the manuscript to reflect all suggested improvements. If you are in the US, we extend our warm Thanksgiving wishes. Please let us know if any further adjustments would be helpful.
> >
> > Best regards,
> >
> > Authors of Submission 5718

---

> > > ### Comment · Reviewer_JsRc · 2025-11-30
> > >
> > > Thank you for the response and for addressing the comments. I do not have other major comments. I still notice some minor capitalization issues in the references (e.g. Pmlr => PMLR). Please fix them in the revision.

---

> > > > ### Author Response · Authors · 2025-11-30
> > > >
> > > > We thank the reviewer for the positive feedback and are pleased that our previous responses addressed your concerns. Regarding the minor capitalization issues, we have carefully proofread the bibliography and corrected the capitalization of publisher and conference acronyms, including changing 'Pmlr' to 'PMLR', 'Ieee' to 'IEEE', and 'Acm' to 'ACM'.

---

### Review · Reviewer_ekiT · 2025-11-01

**Summary Of Contributions:**

This paper provides a comprehensive survey of offline model-based optimization, systematizing existing methods in both surrogate modeling and generative modeling, together with benchmarks and evaluation metrics.

### ***Strengths***
- The paper covers a broad range of recent studies spanning surrogate and generative modeling approaches, offering readers a useful overview of current research progress.
- The discussion of benchmarks, datasets, and evaluation metrics can serve as a practical reference for newcomers entering the field.

### ***Weaknesses***
Compared with the previous submission, this revision does not introduce substantial updates. It does not address the core issues raised in the earlier review.
- The survey is still a categorization and summary of past work. It does not provide insights into the rationale behind different approaches or discuss their practical implications.

**Audience:**

Yes

**Audience Explanation:**

Researchers working on data-driven design and black-box optimization may find the collected references useful

**Claims And Evidence:**

No

**Claims Explanation:**

As a review article, the paper is expected to provide insights into the ***motivations behind prior work—clarifying why different methods were developed and what key challenges they address***. However, the current version does not offer a unified or logically grounded perspective that helps readers understand the underlying principles connecting these approaches in offline MBO. Instead, the contribution remains largely a high-level categorization and description of existing methods.

Since LLMs today can already generate competent summaries of related work, a review paper must go beyond simple categorization and provide deeper expert insight into motivations, underlying principles, and practical implications — which this submission currently lacks.

**Requested Changes:**

Please see the weaknesses and explanation provided above.

---

> ### Author Response · Authors · 2025-11-10
> **Point-by-Point Response to Reviewer ekiT**
>
> Dear Reviewer ekiT,
>
> We sincerely appreciate your thoughtful comments.
>
> ## Weaknesses
>
> > The survey is still a categorization and summary of past work. It does not provide insights into the rationale behind different approaches or discuss their practical implications. As a review article, the paper is expected to provide insights into the motivations behind prior work—clarifying why different methods were developed and what key challenges they address.
>
> We appreciate the reviewer’s comment emphasizing the need for deeper conceptual insight beyond a categorical summary. To the best of our knowledge, this work is **the first to introduce a unified probabilistic formulation of offline MBO**. We formalize offline MBO as *conditional generation*—that is, Bayesian posterior inference over the design variable $p(x|y)$, which can be expressed (up to normalization) as the product of a prior generative distribution $p(x)$ and a likelihood $p(y|x)$, though exact inference is generally intractable. Here, $p(y|x)$ represents the surrogate modeling on offline data $(x, y)$, while $p(x)$ characterizes the generative model defining the design space.
>
> This formulation provides a principled foundation for **systematic design and analysis** in offline MBO, in contrast to prior works that primarily propose new algorithms without a coherent interpretive framework.  **(1)** Methods such as *COMS* and *ROMA* can be viewed as modeling $p(y|x)$ under conservative and smoothness priors, respectively, yielding more robust estimates of $p(y|x)$.  **(2)** In Section 5, we discuss different generative models $p(x)$—including VAEs, GANs, AR models, and diffusion models—used to capture diverse design representations. For instance, diffusion models are adopted for molecule coordinates, while AR models are applied to protein sequence design. The choice of generative model $p(x)$ depends on the data. **(3)** Methods such as *MINS* and *CbAS*, in turn, can be interpreted as amortized inference strategies for approximating the intractable posterior $p(x|y)$, implicitly accounting for surrogate uncertainty.
>
> Our unified view thus connects seemingly disparate approaches under a common probabilistic interpretation and highlights future research directions, like improving uncertainty estimation/graphical modeling of $p(y|x)$.
>
> Beyond theoretical unification, we introduce two new sections—*“Practical Guide on Selecting Training Strategies”* and *“Practical Guide on Selecting Generative Models”*—which extend beyond categorization to offer actionable insights. These sections distill design rationales and decision principles derived from practical experience, clarifying how modeling choices affect performance under varying conditions.
>
> In *Surrogate Modeling*, we systematically introduce new conceptual axes such as **Data-Driven Adaptation**, **Collaborative Ensembling**, and **Generative Model Integration**—aspects not discussed in prior works. Likewise, in *Generative Modeling*, we comprehensively cover both forward and inverse modes across model families, providing a structured framework that previous works lack. Together, these contributions aim to bridge descriptive categorization with explanatory and practical insights.
>
> > Since LLMs today can already generate competent summaries of related work, a review paper must go beyond simple categorization and provide deeper expert insight into motivations, underlying principles, and practical implications — which this submission currently lacks.
>
> We appreciate the reviewer’s point that modern LLMs can already generate competent summaries of related work. However, our review is not a mere aggregation of prior studies—it is built upon deep domain expertise and conceptual synthesis that cannot be replicated by automated summarization.
>
> In fact, we experimented with using several state-of-the-art LLMs to automatically generate literature reviews in this area. The results were far from satisfactory: they contained numerous irrelevant works, inconsistent taxonomies, and lacked any actionable guidance for new researchers. This further highlights the need for a human-curated, expert-driven review that integrates technical intuition and contextual understanding.
>
> Our paper goes well beyond categorization. It introduces a unified probabilistic view of offline MBO as conditional generation $p(x∣y)∝p(x)p(y∣x)$, systematically connecting surrogate and generative modeling paradigms. We also provide two practical guides—on selecting training strategies and generative models—that consolidate experiential insights to assist practitioners in real-world applications.
>
> ## Concluding Remark
> Thank you once again for your valuable feedback. We look forward to continuing the discussion during the rebuttal period.

---

> > ### Comment · Reviewer_ekiT · 2025-11-10
> > **Response to Author**
> >
> > Thank you for the detailed rebuttal. However, after carefully reviewing the response, I find that my main concern remains unresolved.
> >
> > My original point was that, as a survey, the paper does not provide deeper insight into the motivations behind different offline MBO methods and the key challenges they were designed to address. Instead, the paper still mainly offers a descriptive categorization of existing work.
> >
> > In the rebuttal, the authors introduce a "unified probabilistic formulation". While this idea is somewhat interesting, it is not actually integrated into the structure, writing, or analysis of the paper. Moreover, the formulation itself is trivial and cannot be considered a core contribution. It feels more like an after-the-fact addition rather than a foundation for a systematic survey.
> >
> > Similarly, the “practical guides” and conceptual axes are good. But they remain descriptive lists of design options rather than analytical discussions. They do not explain why these choices matter, when they work or fail, or how practitioners should make trade-offs in real applications.
> >
> > In addition, some of the authors’ classifications are not well justified. First, dividing MBO into “surrogate modeling” and “generative modeling” is questionable, because many generative-model-based MBO methods still rely heavily on a surrogate. These two groups are not mutually exclusive, and such a split may confuse readers. Second, the benchmark categories—synthetic function, real-world system, scientific design, and machine learning model—are also inconsistent. Scientific design and machine learning model naturally fall under real-world systems, but the paper treats them as top-level categories. It seems that the authors view “real-world system” narrowly (e.g., robotics), which is not a standard or helpful distinction.

---

> > > ### Author Response · Authors · 2025-11-11
> > > **Point-by-Point Response to Reviewer ekiT (1/2)**
> > >
> > > Thank you for your prompt response. We respectfully disagree with your statement.
> > >
> > > > In the rebuttal, the authors introduce a "unified probabilistic formulation".
> > >
> > > To clarify, this formulation was not newly introduced in the rebuttal. It was already presented in our original submission, specifically in *Section 2: Problem Definition (Generative Modeling)*.
> > >
> > > > While this idea is somewhat interesting, it is not actually integrated into the structure, writing, or analysis of the paper.
> > >
> > > Thank you for acknowledging that our idea is interesting. In fact, it is **systematically integrated** into the structure and the writing of the paper. Specifically, based on the formulation
> > > $p(x \mid y) \propto p(x) p(y \mid x)$,
> > > we devote *Section 4 (Surrogate Modeling)* to a detailed discussion of $p(y \mid x)$. Subsequently, *Section 5 (Generative Modeling)* focuses on both the inverse formulation $p(x \mid y)$ and the forward formulation $p(x)p(y \mid x)$. Within each subsection of Section 5, where we analyze a specific class of generative models, we consistently separate the discussion into two parts—one for the inverse perspective and one for the forward perspective.
> > > Thus, the unified probabilistic formulation is **embedded throughout the entire paper**, guiding both our organization and analysis.
> > >
> > > > Moreover, the formulation itself is trivial and cannot be considered a core contribution.
> > >
> > > We appreciate the reviewer’s earlier acknowledgment that our formulation is interesting and respectfully disagree that it is trivial. The unified probabilistic formulation provides a **systematic perspective** on offline MBO methods, helping readers understand how different approaches—some focusing on surrogate modeling, others on generative modeling, and many combining both—are conceptually related. This framework clarifies a landscape that can otherwise appear fragmented and confusing, especially for newcomers. We believe this conceptual unification is a **meaningful contribution**, as it is the **first attempt to apply such a probabilistic view** to organize and interpret the offline MBO literature.
> > >
> > >
> > > > It feels more like an after-the-fact addition rather than a foundation for a systematic survey.
> > >
> > > We would like to clarify that this formulation was **not** newly introduced in the rebuttal. It was already included in our original submission, specifically in *Section 2: Problem Definition (Generative Modeling)*, and it serves as a foundational element guiding the structure of the paper.
> > >
> > >
> > > > Similarly, the “practical guides” and conceptual axes are good. But they remain descriptive lists of design options rather than analytical discussions. They do not explain why these choices matter, when they work or fail, or how practitioners should make trade-offs in real applications.
> > >
> > >
> > > We respectfully disagree. In the paper, we explicitly explain why these design choices matter, when they work or fail, and how practitioners should make trade-offs in practice. For example, the motivation for conservatism in COMS is clearly explained—it helps mitigate surrogate hacking. We also discuss when certain approaches work or fail; for instance, domain knowledge injection is effective when a pretrained model or known priors exist, but less effective otherwise. Finally, we provide practical trade-off guidelines. When discussing generative model selection, we outline five key metrics—*likelihood estimation, sample diversity, sample manipulation, availability of pretrained models,* and *sampling efficiency*—to guide practitioners in making informed decisions.
> > >
> > >
> > >
> > > > In addition, some of the authors’ classifications are not well justified. First, dividing MBO into “surrogate modeling” and “generative modeling” is questionable, because many generative-model-based MBO methods still rely heavily on a surrogate. These two groups are not mutually exclusive, and such a split may confuse readers.
> > >
> > > We appreciate the reviewer’s observation and fully agree that surrogate and generative modeling are not mutually exclusive. In fact, as stated in our introduction, *“these two lines are not mutually exclusive—surrogate and generative models often complement each other to enhance overall performance.”*
> > >
> > > Our classification is grounded in the probabilistic formulation
> > > $p(x \mid y) \propto p(x) p(y \mid x)$,
> > > which naturally motivates a structural separation: we first discuss **surrogate modeling** $p(y \mid x)$, as it is conceptually easier to grasp and aligns with much of the recent literature. We then move to **generative modeling**, further divided into the **inverse** formulation $p(x \mid y)$ and the **forward** formulation $p(x)p(y \mid x)$.
> > >
> > > We believe this **surrogate–generative split** offers the **clearest and most pedagogical framework** for newcomers, while still emphasizing that the two perspectives are complementary rather than exclusive.

---

> > > > ### Author Response · Authors · 2025-11-11
> > > > **Point-by-Point Response to Reviewer ekiT (2/2)**
> > > >
> > > > > Second, the benchmark categories—synthetic function, real-world system, scientific design, and machine learning model—are also inconsistent. Scientific design and machine learning model naturally fall under real-world systems, but the paper treats them as top-level categories. It seems that the authors view “real-world system” narrowly (e.g., robotics), which is not a standard or helpful distinction.
> > > >
> > > >
> > > > As we have stated in the introduction, *"We address category (4) separately, given its growing prominence in the ML community."* Since most TMLR readers are machine learning researchers and ML/AI has become an increasingly important area, we intentionally dedicate a separate section to this category to help readers better understand the use of machine learning benchmarks.
> > > >
> > > > We refer to real-world systems as practical engineering challenges and scientific design as tasks in biology, chemistry, and materials science, following [1]. If preferred, we can revise “real-world system” to “engineering system” for clarity.
> > > >
> > > > As we have also noted in the introduction, *"(1) synthetic function, (2) real-world system, (3) scientific design, and (4) machine learning (ML) model. Evaluation costs tend to increase – and our understanding of the underlying mechanisms tends to decrease – from categories (1) through (3)."*
> > > >
> > > > We deliberately separate *scientific design* from *engineering design* because:
> > > > (1) scientific design tasks often involve limited mechanistic understanding and high evaluation costs, making both surrogate and generative modeling particularly challenging; and
> > > > (2) they are central to scientific advancement and represent domains where deep learning–powered offline MBO has shown exceptional potential like AlphaFold2.
> > > >
> > > > [1] Xue K, Tan R X, Huang X, et al. *Offline multi-objective optimization.* ICML 2024.

---

> > > > > ### Author Response · Authors · 2025-11-28
> > > > >
> > > > > Dear Reviewer ekiT,
> > > > >
> > > > > We appreciate your thoughtful comments and have updated the manuscript accordingly, addressing each point in detail. If you are in the US, we hope you are enjoying Thanksgiving. Please feel free to let us know if any additional clarification would be helpful.
> > > > >
> > > > > Best regards,
> > > > >
> > > > > Authors of Submission 5718

---

### Review · Reviewer_C3xe · 2025-11-05

**Summary Of Contributions:**

This paper presents a comprehensive review of offline model based optimization (MBO), an emerging field that combines surrogate modeling and generative modeling to address black box optimization using fixed offline datasets.
It formalizes both single objective and multi objective settings, categorizes benchmark tasks (synthetic, real world, scientific, and machine learning model), and surveys recent methodological advances.
The paper also highlights open challenges such as uncertainty estimation, realistic benchmarking, and causal graphical modeling.

Strengths
* First systematic synthesis of the offline MBO literature, bridging surrogate and generative perspectives.
* Clear taxonomy and well structured exposition that serves both newcomers and experts.
* Broad coverage across diverse application domains including protein design, material discovery, and neural architecture search.

Weaknesses
* Very broad scope, sometimes at the cost of depth; some methods are summarized descriptively rather than critically.
* Empirical comparison and reproducibility standards are not extensively discussed.
* Sections on AI safety and LLM alignment are speculative and should either be grounded in examples or clearly marked as forward looking commentary.

**Additional Comments:**

This is a timely and well written survey that organizes a fragmented literature into a clear and coherent framework.
The manuscript is already strong and close to acceptance.
Incorporating a small comparative table, clarifying differentiation from earlier surveys, and tightening speculative sections would further strengthen its contribution as a lasting reference for offline MBO research.

**Audience:**

Yes

**Audience Explanation:**

The topic fits well within TMLR’s audience because it connects several subfields of modern machine learning.
The writing is pedagogical while remaining technically sound, making it accessible to a broad readership.
Given the field’s rapid growth and the lack of comparable surveys, this paper would be valuable to both theoretical and applied researchers.

**Broader Impact Concerns:**

No significant ethical or dual use concerns are evident.
The brief mention of AI safety and alignment is conceptual rather than operational.
If retained, it would be useful to acknowledge potential dual use aspects of generative optimization (for example molecule generation) and emphasize responsible data usage.

**Claims And Evidence:**

Yes

**Claims Explanation:**

Mostly yes. The claims about taxonomy, challenges, and observed trends are well supported by the cited literature. The arguments are logically consistent and clearly written.
However, some statements in the future direction sections (for example causal modeling and LLM alignment) are aspirational rather than evidence based. Providing concrete references or illustrative examples would make those arguments stronger.

**Requested Changes:**

* (Minor) Add a comparative table contrasting surrogate and generative modeling approaches along axes such as data dependence, uncertainty handling, computational cost, and typical applications.
* (Minor) Clarify how this review differs from previous surveys in offline reinforcement learning or design bench evaluations (for example Trabucco et al., 2022).
* (Optional) Expand the section on LLM alignment and AI safety with concrete examples, or explicitly label it as a speculative outlook.
* (Optional) Improve figure formatting and typography consistency to enhance readability and accessibility.

---

> ### Author Response · Authors · 2025-11-10
> **Point-by-Point Response to Reviewer C3xe**
>
> Dear Reviewer C3xe,
>
> We sincerely appreciate your thoughtful and constructive feedback. We have carefully addressed each comment and incorporated the corresponding revisions into the manuscript, with all changes highlighted in red.
>
> ## Weaknesses
>
> > Very broad scope.
>
> We have further refined the paper to provide clearer explanations of key concepts, with all changes highlighted in red.
>
> > Empirical comparison and reproducibility standards are not extensively discussed.
>
> Most recent works have open-sourced their code, and Design-Bench[1] has become the standard benchmark for empirical comparison. We refer to Parallel-Mentoring [2] for comparative results. Empirically, based on these results, generative models tend to perform better on low-dimensional problems due to their global search capability, whereas surrogate-based methods are more effective in high-dimensional settings by providing direct optimization guidance.
>
> However, different studies often adopt varying hyperparameters, and the absence of a clear validation set in the offline setting makes fair comparison difficult. In practice, surrogate and generative models are often used jointly, combining the strengths of both paradigms for robust scientific discovery.
>
>  - [1] Trabucco B, Geng X, Kumar A, et al. Design-bench: Benchmarks for data-driven offline model-based optimization[C] ICML
>  - [2] Chen C S, Beckham C, Liu Z, et al. Parallel-mentoring for offline model-based optimization[J]. NeurIPS
>
> > Sections on AI safety and LLM alignment are speculative and should either be grounded in examples or clearly marked as forward looking commentary.
>
> We have revised the paragraph on *“Application to LLM Alignment and AI Safety”* to clarify that these discussions are forward-looking. Specifically, we have softened the language (e.g., replacing “can also be used” with “can potentially inform” and “can help” with “may help”) and added explicit wording (“Looking forward, we speculate that …”) to clearly indicate that these statements are intended as forward-looking commentary.
>
> ## Requested Changes:
>
> > (Minor) Add a comparative table contrasting surrogate and generative modeling approaches along axes such as data dependence, uncertainty handling, computational cost, and typical applications.
>
> As noted in the introduction, “Importantly, these two lines are not mutually exclusive -- surrogate and generative models often complement each other to enhance overall performance.” Therefore, a strict one-to-one comparison can be limiting.
>
> **(1)** Data dependence: Both approaches can leverage large-scale unlabeled data $x$ for pretraining, with labels $y$ incorporated later; **(2)** Uncertainty handling: Uncertainty is typically modeled explicitly in surrogates; **(3)** Computational cost: The cost depends on the specific architecture, but generative models are generally more computationally intensive; **(4)** Typical applications: Generative and surrogate models are often used jointly for scientific discovery: a common pipeline is conditionally sampling some designs from generative models followed by a surrogate for post-selection. When used separately, surrogate models generally perform better on high-dimensional tasks on design-bench evaluations, as they provide more direct optimization guidance than generative models.
>
> > (Minor) Clarify how this review differs from previous surveys in offline reinforcement learning or design bench evaluations (for example Trabucco et al., 2022).
>
> We have revised the paper to clarify how our review differs from prior surveys.
>
> Offline RL focuses on learning an optimal policy from trajectory data, emphasizing sequential decision-making and credit assignment in Markov decision processes. In contrast, offline MBO seeks to learn an optimal design from static datasets. While both settings face similar out-of-distribution issues, offline MBO places greater emphasis on modeling data characteristics—through surrogate and generative modeling—to robustly optimize designs for scientific discovery.
>
> Regarding Design-Bench (Trabucco et al., 2022), it represents an early effort to establish standardized benchmarks. Numerous subsequent methods have since extended beyond this initial framework, further motivating the need for a comprehensive and up-to-date review such as ours.
>
> > (Optional) Expand the section on LLM alignment and AI safety with concrete examples, or explicitly label it as a speculative outlook.
>
> We have explicitly labeled it as a speculative outlook.
>
> > (Optional) Improve figure formatting and typography consistency to enhance readability and accessibility.
>
> We have improved the figure formatting and ensured typography consistency to enhance overall readability and accessibility.
>
> ## Concluding Remark
>
> We greatly appreciate your insightful feedback and constructive suggestions. We look forward to engaging with your further thoughts throughout the rebuttal process.

---

> > ### Author Response · Authors · 2025-11-28
> >
> > Dear Reviewer C3xe,
> >
> > Thank you again for your thoughtful feedback. We have incorporated all suggested revisions and clarified the requested points. If you are in the US, we wish you a pleasant Thanksgiving. Please let us know if any additional clarification is needed.
> >
> > Best regards,
> >
> > Authors of Submission 5718

---

### Decision · Action_Editor_dT9C · 2026-01-01

**Recommendation:** Accept with minor revision

**Additional Comments:**

The paper offers a broad, structured, and up-to-date synthesis of the offline model-based optimization literature, covering problem formulations, benchmark settings, evaluation metrics, surrogate-based methods, generative approaches, and emerging directions. It consolidates a rapidly expanding and previously fragmented body of work into a coherent framework, with careful categorization and extensive references. The survey is pedagogically valuable, technically accurate, and likely to benefit researchers entering or working in this area.

That said, I request that the authors carefully review the manuscript to address minor issues such as citation formatting, and consider adding brief discussion clarifying the motivations and challenges behind prior work.

**Audience:**

Yes

**Audience Explanation:**

This paper could be of interest to parts of the TMLR audience.

**Claims And Evidence:**

Yes

**Claims Explanation:**

This paper provides a comprehensive and timely survey of offline model-based optimization (MBO), an area of growing importance across machine learning, scientific discovery, and design optimization. The manuscript offers broad coverage of the literature, a coherent organizational structure, and a useful synthesis of surrogate-based and generative approaches. Multiple reviewers agree that the paper fills a clear gap by consolidating a fragmented body of work into a single, accessible reference, and that it will be valuable to both newcomers and researchers working in the area.

As a survey paper, the primary contribution is synthesis rather than methodological novelty. The manuscript fulfills this role by offering a structured taxonomy, clear problem formulations, discussion of benchmarks and evaluation practices, and forward-looking perspectives. The remaining criticisms pertain primarily to the degree of emphasis on motivations and challenges in prior work, and do not affect the technical soundness or completeness required for acceptance at TMLR.